# Dynamic fibronectin assembly and remodeling by leader neural crest cells prevents jamming in collective cell migration

William Duncan Martinson[1], Rebecca McLennan[2], Jessica M Teddy[3], Mary C McKinney[3], Lance A Davidson[4], Ruth E Baker[1], Helen M Byrne[1], Paul M Kulesa[3,5,6], Philip K Maini[1]*

[1]Wolfson Centre for Mathematical Biology, Oxford University, Oxford, United Kingdom; [2]Children's Mercy Kansas City, Kansas City, United States; [3]Stowers Institute for Medical Research, Kansas City, United States; [4]Swanson School of Engineering, University of Pittsburgh, Pittsburgh, United States; [5]Department of Anatomy and Cell Biology, University of Kansas School of Medicine, Kansas City, United States; [6]Department of Biological Sciences, University of Notre Dame, Notre Dame, United States

*For correspondence:
Philip.Maini@maths.ox.ac.uk

**Competing interest:** The authors declare that no competing interests exist.

**Abstract** Collective cell migration plays an essential role in vertebrate development, yet the extent to which dynamically changing microenvironments influence this phenomenon remains unclear. Observations of the distribution of the extracellular matrix (ECM) component fibronectin during the migration of loosely connected neural crest cells (NCCs) lead us to hypothesize that NCC remodeling of an initially punctate ECM creates a scaffold for trailing cells, enabling them to form robust and coherent stream patterns. We evaluate this idea in a theoretical setting by developing an individual-based computational model that incorporates reciprocal interactions between NCCs and their ECM. ECM remodeling, haptotaxis, contact guidance, and cell-cell repulsion are sufficient for cells to establish streams in silico, however, additional mechanisms, such as chemotaxis, are required to consistently guide cells along the correct target corridor. Further model investigations imply that contact guidance and differential cell-cell repulsion between leader and follower cells are key contributors to robust collective cell migration by preventing stream breakage. Global sensitivity analysis and simulated gain- and loss-of-function experiments suggest that long-distance migration without jamming is most likely to occur when leading cells specialize in creating ECM fibers, and trailing cells specialize in responding to environmental cues by upregulating mechanisms such as contact guidance.

## Editor's evaluation

This important study presents predictions from a computational model demonstrating the impact of the extracellular matrix on collective cell migration in the neural crest. The evidence supporting the claims of the authors is solid, and the study is interesting to cell biologists exploring cell migration in different contexts.

## Introduction

Vertebrate neural crest cells (NCCs) are an important model for collective cell migration. Discrete streams of migratory NCCs distribute throughout the embryo to contribute to nearly every organ (*Le Douarin and Kalcheim, 1999*; *Tang and Bronner, 2020*). Consequently, mistakes in NCC migration

may result in severe birth defects, termed neurocristopathies (*Vega-Lopez et al., 2018*). In contrast to well-studied tightly adhered cell cluster models, such as border cell and lateral line migration (*Peercy and Starz-Gaiano, 2020*; *Olson and Nechiporuk, 2021*), much less is known about how 'loosely' connected cells, such as NCCs, invade through immature extracellular matrix (ECM) and communicate with neighbors to migrate collectively. Moreover, several invasive mesenchymal cancers resemble collective NCC migration, with cells moving away from the tumor mass in chain-like arrays or narrow strands (*Friedl et al., 2004*; *Friedl and Alexander, 2011*). Therefore, utilizing the neural crest experimental model to gain a deeper knowledge of the cellular and molecular mechanisms underlying collective cell migration has the potential to improve the repair of human birth defects and inform strategies for controlling cancer cell invasion and metastasis.

Time-lapse analyses of NCC migratory behaviors in several embryo model organisms have revealed that leader NCCs in discrete migratory streams in the head (*Teddy and Kulesa, 2004*; *Genuth et al., 2018*), gut (*Druckenbrod and Epstein, 2007*; *Young et al., 2014*), and trunk (*Kasemeier-Kulesa et al., 2005*; *Li et al., 2019*) are highly exploratory. Leading NCCs extend thin filopodial protrusions in many directions, interact with the ECM and other cell types, then select a preferred direction for invasion. Trailing cells extend protrusions to contact leaders and other cells to maintain cell neighbor relationships and move collectively (*Kulesa et al., 2008*; *Ridenour et al., 2014*; *Richardson et al., 2016*). These observations suggest a leader-follower model for NCC migration (*McLennan et al., 2012*) in which NCCs at the front of migrating collectives read out guidance signals and communicate over long distances with trailing cells. This model challenged other proposals that NCCs respond to a chemical gradient signal and sustain cohesive movement via an interplay of local cell co-attraction and contact inhibition of locomotion (*Theveneau and Mayor, 2012*; *Szabó et al., 2016*; *Merchant et al., 2018*; *Merchant and Feng, 2020*). The precise cellular mechanisms that underlie leader-follower migration behavior, for example, the involvement with ECM or the nature of communication signals created by leaders, have not been elucidated.

One paradigm that has emerged suggests that loosely connected streams of NCCs move in a collective manner by leaders communicating through long-range signals that are interpreted and amplified by follower NCCs. In support of this, chick leader cranial NCCs show enhanced expression of secreted factors, such as angiopoietin-2 (Ang-2) (*McLennan et al., 2015a*) and fibronectin (FN) (*Morrison et al., 2017*). Knockdown of Ang-2 reduces the chemokinetic behavior of follower chick NCCs, resulting in disrupted collective cell migration (*McKinney et al., 2016*). These observations suggest that follower NCC collective movement depends on signals deposited in the microenvironment by leaders or the adjacent mesoderm.

Signals in the FN-rich ECM may also provide microenvironmental cues for NCC collective migration. NCCs cultured in FN-rich ECM move with thin, anchored filopodia through a field that is punctate immediately distal to the migrating front, while 'pioneer' NCCs within the front appear to form radially oriented bundles of FN-containing filaments (*Rovasio et al., 1983*). Later work on the NCC epithelial-to-mesenchymal transition showed that inhibition of ECM-integrin receptors resulted in delamination of NCCs into the neural tube lumen (*Kil et al., 1996*). Moreover, FN is abundant in the mouse head and neck (*Mittal et al., 2010*), and there is a dramatic reduction in the number of migrating NCCs that reach the heart in FN null mice (*Wang and Astrof, 2016*). Thus, FN is critical for NCC to reach their targets, although its precise role in collective cell migration remains unclear.

Mathematical modeling of NCC migration can help provide insight into the role of FN and identify the origins of multiscale collective behaviors (*Giniūnaitė et al., 2020*; *Kulesa et al., 2021*). Previous models for NCC migration demonstrated that contact guidance, a mechanism by which cells align themselves along ECM fibers, can establish collective behavior and create single-file cell chains (*Painter, 2009*). Later models incorporating ECM degradation and haptotaxis, a process by which cells migrate up adhesive ECM gradients, demonstrated how matrix heterogeneity and cell-cell interactions could determine cell migratory patterns (*Painter et al., 2010*; *Wynn et al., 2013*). Models for other collectively migrating cells, such as fibroblasts, suggest that ECM remodeling by motile cells can enable anisotropic collective movement, with migratory directions dictated by fiber orientation (*Dallon et al., 1999*; *Groh and Louis, 2010*; *Chauviere et al., 2010*; *Azimzade et al., 2019*; *Wershof et al., 2019*; *Pramanik et al., 2021*; *Suveges et al., 2021*; *Metzcar et al., 2022*). No models have yet addressed how collective migration arises from cell remodeling of an initially isotropic, immature ECM.

In this paper, we develop an agent-based model (ABM; also known as an individual-based model) of chick cranial NCC migration that considers a cell-reinforced migratory cue in which 'leader' cells remodel an initially punctate and immature FN matrix to signal 'follower' cells. Other processes detailed in the model include cell-cell repulsion, haptotaxis, and contact guidance. This framework incorporates new observations of FN distribution in the chick cranial NCC microenvironment and simulates gain- and loss-of-function of FN. Detailed simulations of the model over parameter space, coupled with global sensitivity analysis of ABM parameters, identifies mechanisms that dominate the formation of migrating streams and other NCC macroscopic behavior. Through this analysis, we find that migration is most efficient when leading NCCs specialize in remodeling FN to steer the collective, as cells otherwise enter a 'jammed' state in which migration is greatly reduced and cells are densely packed together (*Sadati et al., 2013*). The addition of 'guiding signals' that direct NCCs along a target corridor limits excessive lateral migration in the ABM but concurrently promotes separation between leader and trailing cells in the stream. We survey ABM parameter combinations that recover robust collective migration without such stream breaks, which underscore the potential importance of contact guidance.

## Results

### FN protein expression within the mesoderm is unorganized and punctate prior to cranial NCC emigration, but filamentous after it has been traversed by leader cells

We confirmed that FN throughout the head, neck and cardiovascular regions in the chick embryo is distributed in patterns that overlap with NCC migratory pathways (*Figure 1*; *Duband and Thiery, 1982*). To assess the in vivo distribution of FN protein at higher spatial resolution, we examined transverse cryosections (*Figure 1A*; *Hamburger and Hamilation, 1951*) at developmental stages (HH12–13) midway through cranial NCC migration to reveal that NCCs are in close association with a heterogeneous meshwork of FN (*Figure 1B*). Punctate FN, not yet fibrils, are found both distal to the invasive NCC migratory front and adjacent to the leader NCC subpopulation (*Figure 1C*). By contrast, elongated FN fibrils are located behind the leading edge of invasive NCC streams (*Figure 1D*). FN fibrils proximal to the invasive NCC migratory front did not reveal a preferred directional orientation. Similar punctate FN was also visible in regions outside the NCC migratory pathway, subjacent to the surface ectoderm (*Figure 1C*).

### Gain- or loss-of-function of FN leads to reduced NCC migration

Functional analysis confirms FN is required for normal NCC migration. Knockdown of FN expression introduced into premigratory NCCs within the neural tube led to a significant reduction (nearly 30%) in the total distance migrated by NCCs enroute to BA2 (*Figure 1E*). Microinjection of soluble FN into the cranial NCC migratory domain, for example, into the paraxial mesoderm adjacent to rhombomere 4 (r4) prior to NCC delamination, also led to a dramatic reduction (by 70%) in NCC migration as indicated by a decrease in the area typically covered by the invading NCCs (*Figure 1F*). Thus, decreasing the expression of FN – presumably by decreasing the rate at which FN is secreted by motile NCCs – or increasing the FN density in the microenvironment led to significant changes in the NCC migration pattern. We conclude that a balance of FN within the migratory microenvironment is required to promote proper migration.

### An individual-based model of NCC migration and ECM remodeling produces collectively migrating streams in silico

Our observations led us to hypothesize that migrating NCCs remodel punctate FN into a fibrous scaffold for trailing cells. We evaluated this idea in a theoretical setting by constructing a mathematical model in which NCCs are represented as discrete off-lattice point masses, that is, agents, that freely move in a two-dimensional (2D) plane (*Figure 2—figure supplement 1*). Each agent responds to, and influences, the remodeling of an initially punctate ECM (*Figure 2*; *Figure 2—figure supplement 2*). Their velocities are determined using an overdamped version of Newton's second law. The three types of forces that alter agent trajectories arise from friction (which is proportional to the cell velocity), cell-ECM interactions (specifically, those from haptotaxis and contact guidance), and cell-cell

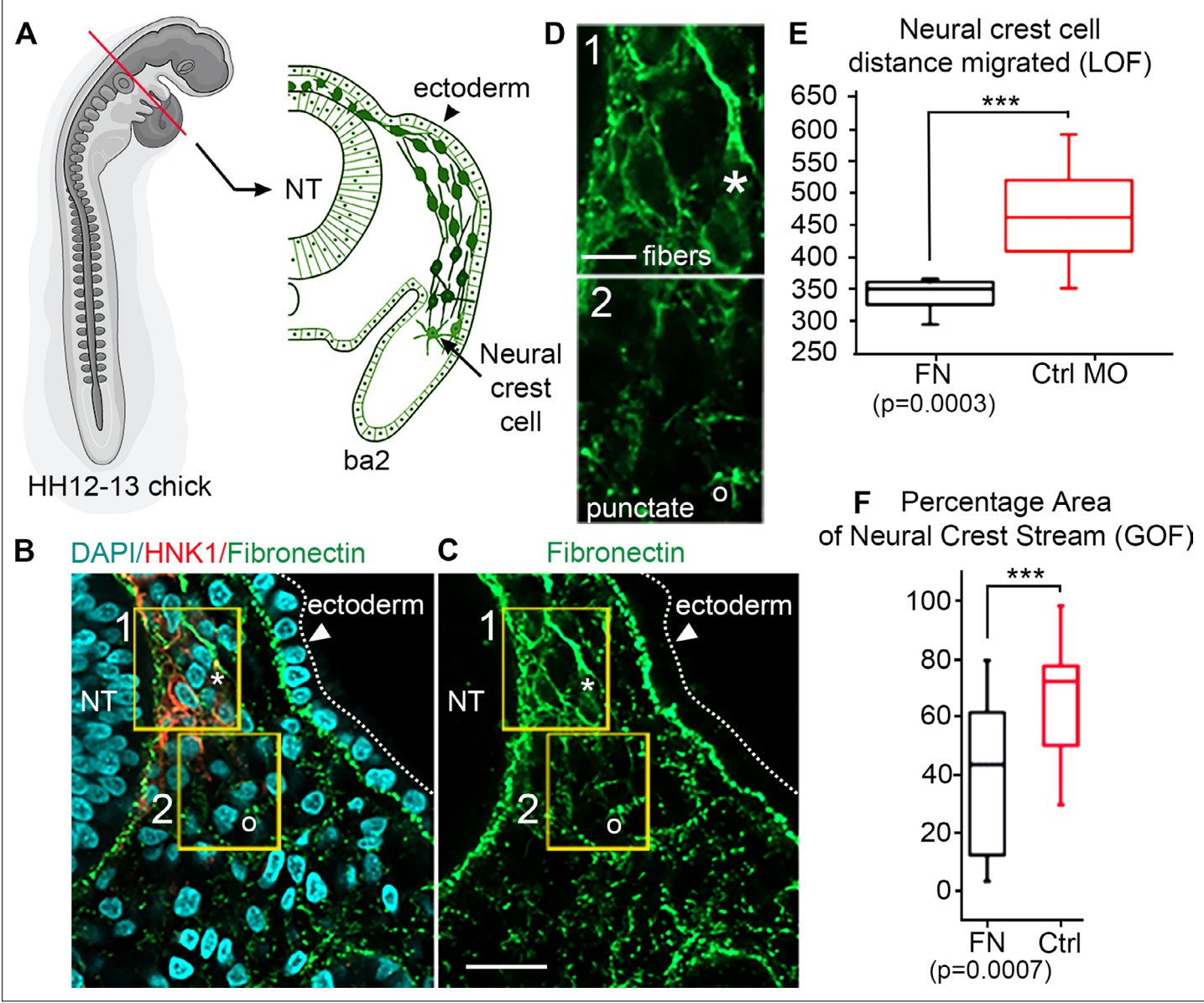

**Figure 1.** In vivo observations of fibronectin (FN) and results from gain/loss-of-function experiments. (**A**) Schematic of a typical neural crest cell (NCC) migratory stream in the vertebrate head of the chick embryo, at developmental stage HH12–13 (*Hamburger and Hamilation, 1951*) at the axial level of the second branchial arch (ba2). (**B**) Transverse section through the NCC migratory stream at the axial level of rhombomere 4 (**r4**) triple-labeled for FN (green), nuclei (DAPI-blue) and migrating NCCs (HNK1-red). NCCs migrate subjacent to the surface ectoderm after emerging from the dorsal neural tube (NT). The arrowhead points to the surface ectoderm. The yellow boxes highlight the tissue subregions that contain the leader NCCs (box 1) and are distal to the leader NCCs (box 2), marking distinct shapes of the FN in each box with an asterisk (box 1) and open circle (box 2). (**C**) FN only. (**D**) The fibrous (box 1-asterisk) and punctate (box 2-open circle) appearance of FN in the NCC microenvironment. (**E**) NCC distance migrated in FN morpholino-injected embryos and (**F**) percentage area of the NCC migratory stream after microinjection of soluble FN into the r4 paraxial mesoderm prior to NCC emigration. NT = neural tube. The scale bars are 50 μm (**B–C**) and 10 μm (**D**).

The online version of this article includes the following figure supplement(s) for figure 1:

**Figure supplement 1.** Morpholino knockdown of fibronectin in cranial neural crest cells.

repulsion. Neighboring NCCs, FN puncta, and FN fibers generate the latter two forces and affect the motion of an agent only when they are within a user-specified distance, $R_{filo}$ , of the agent center (this distance represents the length of cell filopodia protrusions).

The cell-repulsion (resp. cell-ECM) force accounts for the collective interactions that an agent experiences from neighboring NCCs (resp. FN puncta and fibers). The magnitude of the total cell-cell

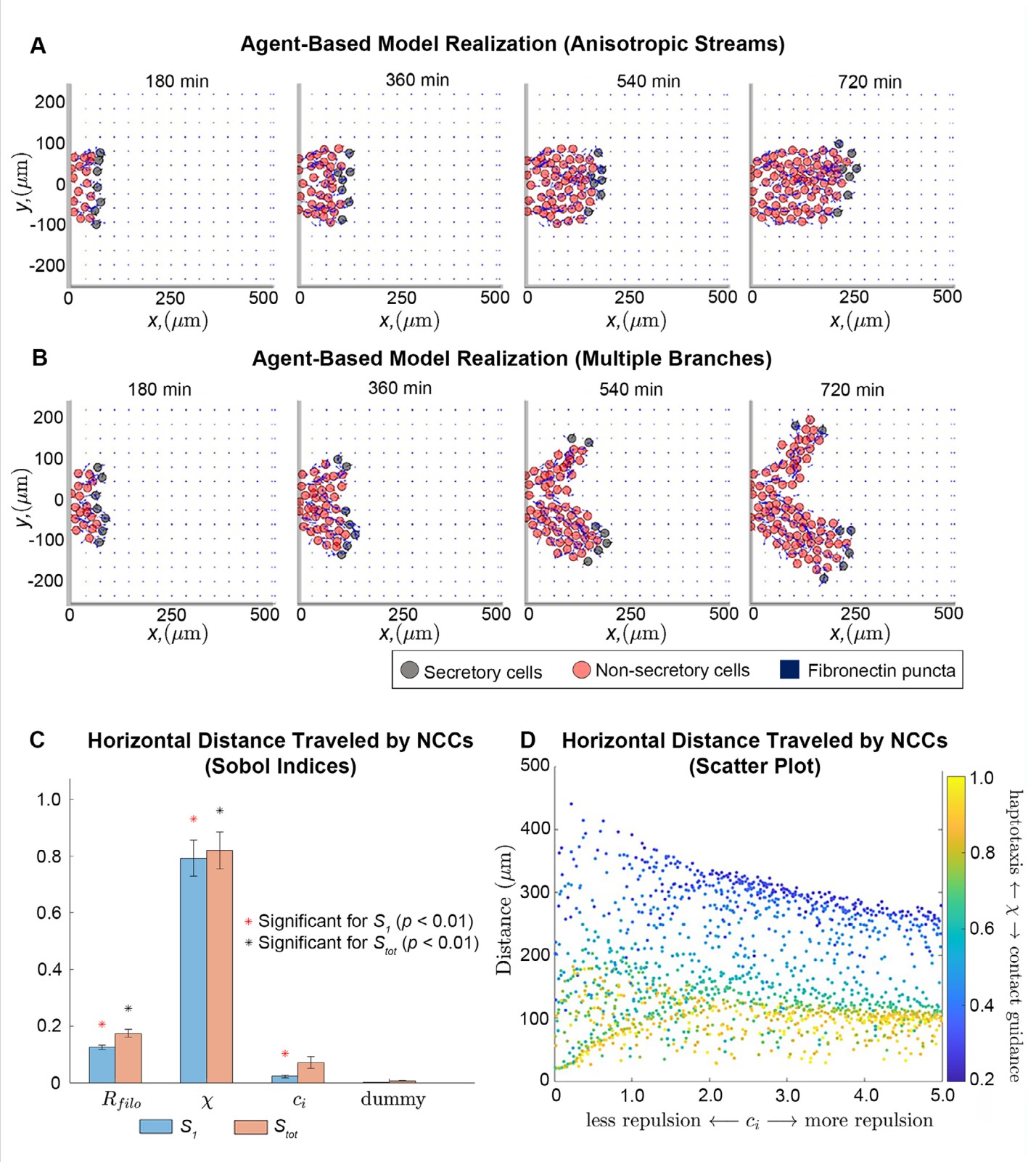

**Figure 2.** Integration of extracellular matrix (ECM) within an agent-based model (ABM) of neural crest cell (NCC) migration. Individual snapshots (**A**) of an example ABM realization reveals that the model can generate a single anisotropically migrating stream over a simulated time of 12 hr. Different realizations generated using the same parameter values but different random seeds (**B**), however, may produce streams that exhibit the formation of multiple branches and demonstrate the range of possible model behaviors. Black circles denote leader cells, which can secrete new FN, red cells signify

*Figure 2 continued on next page*

*Figure 2 continued*

follower non-secretory cells, and blue squares correspond to FN puncta. Arrows denote cell velocities or fiber orientations. The extra column of FN at the right boundary is an artifact of visualization. Sobol indices (**C**) and scatter plots (**D**) of the horizontal distance traveled by NCCs indicate that this metric is most sensitive to the haptotaxis-contact guidance weight, $\chi$, and the filopodial length, $R_{filo}$, but is less dependent on parameters related to cell-cell repulsion ($c_i$). Statistical significance in (**C**) is determined from a two-sample $t$-test that compares the Sobol indices produced by the parameter of interest to those obtained from a dummy parameter (red asterisks indicate significant first-order indices, black asterisks indicate significant total order indices, p<0.01). Each data point in (**D**) represents the average of 20 ABM realizations.

The online version of this article includes the following video and figure supplement(s) for figure 2:

**Figure supplement 1.** Overview of the agent-based model (ABM).

**Figure supplement 2.** Overview of fibronectin (FN) remodeling in the agent-based model.

**Figure supplement 3.** In silico fibronectin distributions in individual realizations.

**Figure supplement 4.** Average fibronectin fiber orientation in global parameter sweep.

**Figure supplement 5.** The nearest neighbor distance is affected by parameter values related to cell-cell repulsion and contact guidance.

**Figure supplement 6.** The migration of cells lateral to the target corridor is largely determined by how they respond to the extracellular matrix (ECM).

**Figure supplement 7.** The nearest neighbor distance and the distance that cells travel are correlated.

**Figure supplement 8.** Histogram of the number of trailing cells that enter the domain.

**Figure supplement 9.** Bar charts of partial rank correlation coefficient (PRCC) values for (**A**) the maximum distance neural crest cells (NCCs) travel in the horizontal direction, (**B**) the maximum range of NCCs in the vertical direction, and (**C**) average nearest neighbor distance, using a larger set of parameter values than the ones considered in the main text.

**Figure 2—video 1.** An example agent-based model (ABM) simulation corresponding to *Figure 2A* of the main text.
https://elifesciences.org/articles/83792/figures#fig2video1

**Figure 2—video 2.** An example agent-based model ( ABM) simulation corresponding to *Figure 2B* of the main text.
https://elifesciences.org/articles/83792/figures#fig2video2

**Figure 2—video 3.** An example agent-based model (ABM) simulation showing how only the fibronectin (FN) distribution changes.
https://elifesciences.org/articles/83792/figures#fig2video3

**Figure 2—video 4.** An example agent-based model (ABM) simulation showing how only the fibronectin (FN) distribution changes.
https://elifesciences.org/articles/83792/figures#fig2video4

repulsion force is determined from the sum of radially oriented forces from all neighboring NCCs, whose strength, modulated by a user-defined constant, $c_i$, decreases quadratically with respect to the distance from the agent center of mass. To represent the finding that chick cranial NCCs do not always repel each other upon contact (*Kulesa and Fraser, 1998*; *Kulesa et al., 2004*), we adjust the direction of the resultant force by stochastically sampling from a von Mises distribution (*Mardia and Jupp, 1999*). The parameters of the distribution depend on the number and location of NCCs sensed by the agent. They ensure it is uniform when few NCCs are present but cause it to resemble a periodic normal distribution biased in the direction of lowest NCC density when many cells are sensed (for details, see the Materials and methods section). Thus, when cells sense each other at longer ranges, they may align. When cells are close enough to be overlapping, however, they are more likely to repel each other along the direction of contact, similarly to other implementations of cell-cell repulsion (*Colombi et al., 2020*). We have verified that increasing the magnitude of the cell-cell repulsion forces increases the average nearest neighbor distance between cells in the ABM (*Figure 2—figure supplement 5*; *Figure 3—figure supplement 4*).

The magnitude of the total cell-ECM force is similarly determined by summing radially oriented forces originating from neighboring FN puncta and fibers, with the strengths of such forces decreasing quadratically with respect to distance from the agent center. The resultant cell-ECM force direction is determined by a linear combination of signals arising from haptotaxis and contact guidance, which are weighted by a parameter, $\chi$. The haptotaxis and contact guidance cues are both represented as unit vectors. Haptotaxis biases the total cell-ECM force toward increasing FN densities and its cue is sampled from a von Mises distribution whose parameters depend on the number and location of neighboring FN puncta and fibers sensed by the cell. The distribution is biased such that the cell is likely to travel toward the greatest FN concentration sampled. Contact guidance aligns the cell-ECM force along the direction of FN fibers. The unit vector corresponding to this cue is computed from the average orientation of FN fibers sensed by the agent.

Cells migrate through an initially isotropic, equi-spaced, square lattice of FN puncta that approximates the matrix distribution prior to NCC migration. Leader cells are represented by a fixed number of 'secretory' cells that generate new FN puncta at their centers according to times drawn from an exponential distribution with user-specified mean. The cells start at the left-hand boundary of the lattice, which we take to represent the section of the neural tube located along r4. Non-secretory cells, which cannot create new FN puncta, enter the domain at later times, provided sufficient space is available. We note that the terms 'secretory' and 'non-secretory' refer only to the ability of cells to secrete new FN puncta, as both cell types can create and align fibers emanating from puncta they pass over.

Observations of individual ABM simulations indicate that ECM remodeling, haptotaxis, contact guidance, and cell-cell repulsion can generate stream-like patterns, with secretory cells able to maintain their positions at the front of the stream (*Figure 2A*; *Figure 2—video 1*). Cells can migrate anisotropically, even though the initial FN lattice is isotropic, and trailing cell velocities are oriented toward leader cells (for instance, the average angle that the follower cell velocity vector makes with the horizontal axis is 0.11 radians, or 6 degrees, for the simulation shown in *Figure 2A*). Stream-like patterns leave most ECM fibers aligned parallel to the horizontal axis, that is, the direction of the target corridor (*Figure 2—figure supplements 3–4*; *Figure 2—video 3*).

NCCs do not always maintain a single mass as they migrate collectively. In some cases, ABM streams split into two or more 'branches' that colonize regions along the vertical axis (*Figure 2B*; *Figure 2—videos 2 and 4*). Streams split because there is no signal specifically guiding NCCs in a direction parallel to the horizontal axis. Thus, in some cases cells may sense and travel toward FN puncta located perpendicular to the target corridor. We conclude that additional signals are required to prevent NCC stream branching. In later sections, we will investigate how such signals affect the robustness of patterns formed by NCCs.

## Global sensitivity analysis suggests a key role for contact guidance in establishing long-distance NCC migration

To determine how collective migration depends on ABM parameters, we next applied extended Fourier amplitude sensitivity testing (eFAST; *Saltelli et al., 1999*) to examine summary statistics for collective migration. This analysis calculates Sobol indices (*Figure 2C*), which measure the fraction by which the variance of a given summary statistic is attributable to changes in a parameter value of interest (larger values indicate that the statistic is more sensitive to the parameter). The eFAST analysis produces two types of Sobol indices: first-order indices, which record the fraction of variance directly attributable to a parameter of interest, and total-order indices, which include additional synergistic effects that arise when other parameters are altered.

The resulting Sobol indices indicate that the distance NCCs travel in the horizontal direction (i.e. along the target corridor) is most sensitive to $\chi$, the parameter determining the degree to which NCC directional migration in response to the FN matrix is dominated by haptotaxis ($\chi = 1$), contact guidance ($\chi = 0$), or a linear combination of the two. Increasing the value of this parameter decreases the distance cells travel (partial Spearman rank correlation coefficient, PRCC: –0.91, p<0.0001). The cell filopodial radius, $R_{filo}$, supplies the next largest Sobol indices and exhibits a monotonically increasing relationship with the statistic (PRCC: 0.63, p<0.0001). This make sense, as increasing the number of FN puncta and fibers that the cell senses would be expected to generate more persistent and faster migration in the ABM. The parameter for the cell-cell repulsion force strength, $c_i$, presents a small but statistically significant first-order index. Subsequent analysis of its scatter plot (*Figure 2D*) reveals that when contact guidance dominates the cell-ECM force ($\chi < 0.5$), increasing the cell-cell repulsion strength decreases the distance that the streams migrate (PRCC: –0.64, p<0.0001). When haptotaxis dominates the direction of the force ($\chi > 0.8$), this relationship is weaker but remains monotonically decreasing (PRCC: –0.59, p<0.0001). Hence, the distance traveled by NCCs is most sensitive to the mechanism by which cells respond to the ECM, with contact guidance favoring longer streams, while haptotaxis and cell-cell repulsion are negatively correlated with the distance that NCCs travel.

We observed similar results for a summary statistic, the maximum extent of the migrating stream in directions perpendicular to the target corridor, that indicates the lateral spread of cells in the ABM from the target corridor (*Figure 2—figure supplement 6*; we will hereafter refer to this statistic as the 'lateral spread'). The cell-ECM weight, $\chi$, again generates the largest Sobol indices for this statistic

and exhibits a strong negative monotonic relationship (PRCC: –0.90, p<0.0001), indicating that lateral migration increases with upregulation of contact guidance. By contrast, scatter plots suggest this statistic is weakly correlated with the filopodial radius (PRCC: 0.12) and cell-cell repulsion strength (PRCC: –0.08).

Analysis of Sobol indices for the average distance between neighboring cells reveals the importance of haptotaxis and cell-cell repulsion on cell clustering. The cell-ECM weight and the cell-cell repulsion strength parameter both yield significant first-order Sobol indices (*Figure 2—figure supplement 5*), with the latter parameter generating larger values. Only the total-order Sobol index is significant for the filopodial radius, which implies that this parameter affects cell-cell separation largely via its higher-order interactions with the other two parameters. This makes sense, given that the cell-ECM and cell-cell repulsion forces are indirectly affected by changing the number of FN puncta and NCCs sensed. Scatter plots (*Figure 2—figure supplement 5*) suggest a monotonically decreasing relationship between the nearest neighbor distance and the cell-ECM weight, $\chi$ (PRCC: –0.74, p<0.0001), a monotonically increasing relationship with the cell-cell repulsion parameter, $c_i$ (PRCC: 0.78, p<0.0001) and almost no correlation for the filopodial radius (PRCC: 0.05, p=0.07).

## Leading NCCs drive collective migration by secreting and remodeling ECM substrates

To identify combinations of parameters that might regulate collective migration, we simulated experimental manipulations that affect cell synthesis of new FN (*Figure 3A*; see Materials and methods) while leaving cell assembly of fibers from existing puncta unchanged. When leader secretory cells secrete FN puncta more rapidly, we find that the stream travels further, by roughly 18%. By contrast, if neither cell type secretes new FN puncta, then the average distance the stream travels decreases by about 5% (collective migration still occurs because cells can respond to pre-existing FN). Finally, if both cell types secrete FN, such that there is no difference between cell phenotypes, then the average distance migrated by cells decreases from the baseline levels regardless of whether secretion occurs at the baseline rate (21% decrease) or is elevated (19% decrease). Similar results hold for statistics measuring the lateral spread of NCCs (*Figure 3—figure supplement 1*).

We next modulate how cells alter the FN matrix, by eliminating both FN secretion and fiber assembly in leader and/or follower cells (*Figure 3B*). When leading secretory cells (or both cell types) are unable to remodel the matrix, the average distance traveled by the NCC stream decreases by 47% (resp. 51%). When follower 'non-secretory' cells are prevented from assembling and reorienting FN fibers, however, the effect is minimal with only a 2% reduction in distance traveled. Similar results are observed for statistics measuring the nearest neighbor distance and the lateral spread of NCCs (*Figure 3—figure supplement 2*).

These results demonstrate that, within the ABM, FN remodeling by leading NCCs plays a key role in determining long-distance collective migration. By contrast, no discernible differences are observed when trailing cells cannot remodel the matrix. Our findings suggest that collective migration may be made more effective when leading and trailing NCCs perform specialized roles, with leading cells remodeling the ECM.

## The initial FN matrix distribution is crucial to establishing long-distance collective cell migration

ECM remodeling can direct collective cell migration, but the degree to which the pre-existing ECM affects cell trajectories remains unclear. Increasing the initial lattice spacing between FN puncta from 20 to 60 µm hinders collective migration in the ABM, decreasing the distance that the stream migrates by 36%. The lateral spread of NCCs is similarly reduced within sparser matrices (*Figure 3—figure supplement 3*). We conclude that if NCCs are less likely to sense FN in their local environment, then they are consequently less likely to migrate as far. Additionally, the nearest neighbor distance between cells decreases (*Figure 3—figure supplement 3*). Such an increased cell density and low motility state resembles a 'jamming' scenario in which cells are tightly packed, resistant to movement, and behave like a solid (*Sadati et al., 2013*). These similarities lead us to classify cells in the ABM as being in a 'jammed' state when they collectively travel less than 100 µm over 12 hr, as we have found a moderate but statistically significant positive correlation between the nearest neighbor distance and

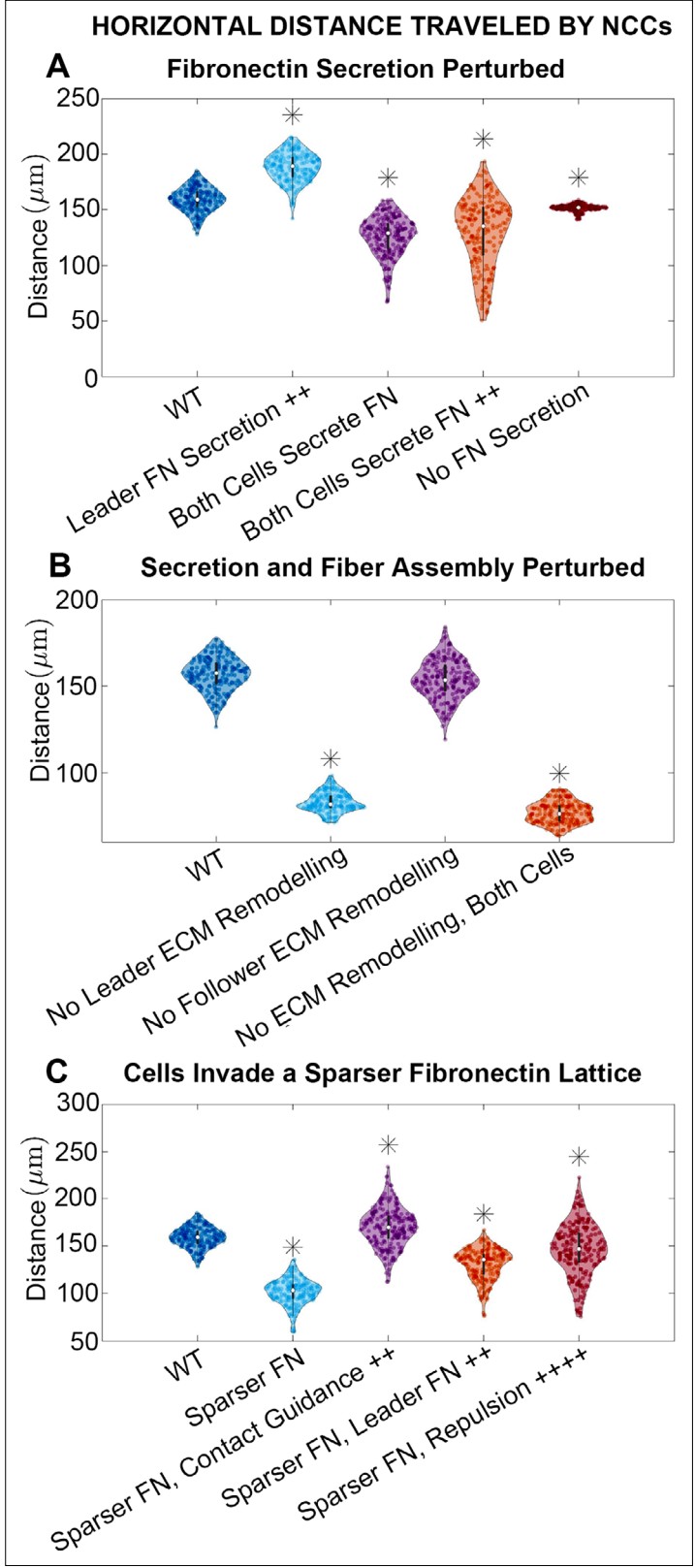

**Figure 3.** Horizontal distance traveled by neural crest cells (NCCs) after 12 hr. Violin plots for experiments in which (**A**) fibronectin (FN) secretion is perturbed, (**B**) secretion and fiber assembly (FN remodeling) are altered, and (**C**) cells invade a sparser (60 μm) FN lattice suggest that the extracellular matrix (ECM) plays an important role in NCC migration. Overexpression experiments (++) decrease the average timescale over which a cell makes new

*Figure 3 continued on next page*

*Figure 3 continued*

FN from 30 to 10 min, respectively. Contact guidance and cell-cell repulsion upregulation in (**C**) correspond to $\chi = 0.33$ and $c_i = 5$, respectively. Asterisks indicate whether distributions are significantly different (p<0.01) from that of the baseline (WT) parameter regime using a Mann-Whitney U-test. Two-hundred agent-based model (ABM) realizations are used to generate each violin plot.

The online version of this article includes the following figure supplement(s) for figure 3:

**Figure supplement 1.** Additional violin plots for in silico knockdown/overexpression experiments of fibronectin (FN).

**Figure supplement 2.** Violin plots showing how (**A**) the maximum range of neural crest cells (NCCs) in the vertical direction and (**B**) average nearest neighbor distance change for the in silico experiments in which fibronectin secretion and fiber assembly (extracellular matrix [ECM] remodeling) are altered in either secretory 'leader' cells or non-secretory 'follower' (see *Figure 3B* of the main text).

**Figure supplement 3.** Violin plots showing how (**A**) the maximum range of neural crest cells (NCCs) in the vertical direction and (**B**) average nearest neighbor distance change for the in silico experiments in which cells invade a sparser (60 μm) fibronectin (FN) lattice (see *Figure 3C* of the main text).

**Figure supplement 4.** Violin plots showing how (**A**) the maximum distance neural crest cells (NCCs) travel in the horizontal direction, (**B**) the maximum range of NCCs in the vertical direction, and (**C**) average nearest neighbor distance change for in silico experiments in which parameters related to cell-cell repulsion, $c_i$, are altered.

the distance traveled in the horizontal direction (*Figure 2—figure supplement 7*; *Figure 3—figure supplement 4*).

Cells can, however, upregulate certain mechanisms to compensate for sparser FN distributions and re-establish long-distance collective migration (*Figure 3C*). If contact guidance is upregulated in the ABM (e.g. if the weight $\chi$ decreases from 0.5 to 0.33), then the NCC migration distance in sparse FN increases by 67%. The distance exceeds that obtained in the denser (20 μm) FN lattice by 7%. Increasing the cell-cell repulsion strength or the rate at which secretory cells produce FN also cause NCCs to travel greater distances within the sparser FN matrix (by 44% and 28%, respectively), but these changes do not rescue migration to distances comparable to those in denser environments. These results highlight the importance of contact guidance, ECM remodeling, and/or cell-cell repulsion in rescuing long-distance migration, even when cells are in a jammed state where migration is otherwise hindered.

## Adding directional guidance correctly steers cells along their target corridor, but renders the model sensitive to stream breaks and cell separation

As noted above, mechanisms that generate collective migration do not guarantee that cells robustly travel along a single target corridor. Thus far, our ABM simulations have generated NCC streams that typically spread out to widths almost double that of the neural tube corridor from which they emerge. By contrast, in vivo streams typically maintain a constant width from the neural tube and do not branch (*Szabó et al., 2016*; *Kulesa et al., 2004*). Such discrepancies motivate us to introduce a new force (a 'guiding force') that steers cells along the target corridor. This force may represent signals previously neglected in the ABM, such as chemotaxis (*McLennan et al., 2010*; *McLennan et al., 2012*) or domain growth (*Shellard and Mayor, 2019*; *McKinney et al., 2020*). For simplicity, we direct this force along the horizontal axis by introducing a weight, $z$, to control the degree to which the cell velocity is aligned along the corridor track, countering off-axis forces arising from haptotaxis, contact guidance, and cell-cell repulsion. We consider cases for which the guiding force acts on both cell types in the ABM and others for which it only affects leading secretory cells. The second case aims to determine whether leading cells alone can guide the entire migrating collective, even if trailing cells are free to move laterally from the stream.

The guiding force indeed reduces lateral migration from the target corridor as its 'strength', $z$, increases. The NCC stream narrows by as much as 71% when the guiding force acts on both cell types or by 88% when the force only affects leading cells (*Figure 4A*). Cells also migrate longer distances along the corridor (*Figure 4B*). As we expected, the distance migrated is linearly correlated with the guiding force strength, regardless of whether the guiding force acts on leader cells or on both

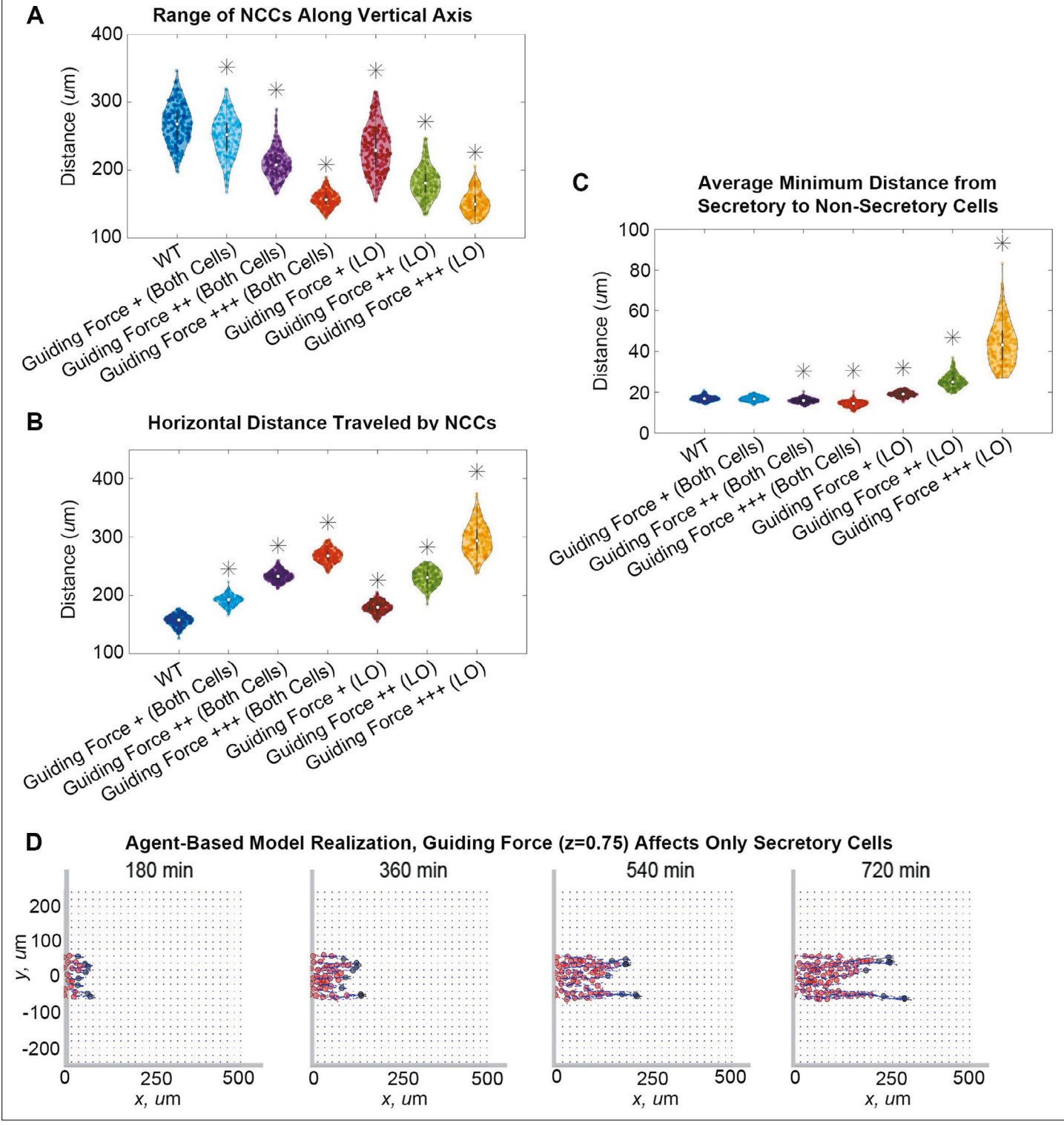

**Figure 4.** Guiding forces reduce lateral spread of a neural crest cell (NCC) stream but can result in stream breakage. Violin plots for (**A**) the range of NCCs along the vertical axis, (**B**) the horizontal distance traveled by NCCs, and (**C**) the average minimum distance from secretory to non-secretory cells in experiments when the guiding force strength, $z$, changes (guiding force +/++/+++: $z = 0.25$ / $z = 0.5$ / $z = 0.75$) in both cell types or only leader secretory cells (LO). Asterisks indicate significantly different (p<0.01) distributions compared to the baseline (WT, $z = 0$) parameter regime according to a Mann-Whitney U-test. Snapshots in (**D**) present the results of an agent-based model (ABM) realization over 12 hr when the guiding force ($z = 0.75$) affects only secretory cells. Two-hundred ABM realizations are used to generate each violin plot.

The online version of this article includes the following video for figure 4:

*Figure 4 continued on next page*

*Figure 4 continued*

**Figure 4—video 1.** An example agent-based model (ABM) simulation corresponding to *Figure 4D* of the main text.
https://elifesciences.org/articles/83792/figures#fig4video1

cell types (leader-only Pearson correlation coefficient: 0.9793, p=0.02; both cell types PCC: 0.9993, p=0.007).

Improvements in the NCC stream trajectory, however, can also drive leader secretory cells to separate from non-secretory follower cells if the former are the only cell type to experience guiding forces. In such cases, the distance between leader and follower cells (*Figure 4C*) increases with the guiding force strength by up to 160%, indicating fragmentation of the NCC stream. However, no breaks occur when the guiding force acts on both cell types, as the distance between leader and follower cells decreases slightly with the guiding force strength (by about 15% from its baseline value). Individual simulations (*Figure 4D*; *Figure 4—video 1*) also show NCC streams can exhibit chain-like migration, with two or more branches forming that are one to two cells in width. Such chains still follow paths of remodeled FN fibers created by leader cells, even when leaders and followers become physically separated. This observation supports the idea that remodeled ECM can provide a long-range cue for follower cells.

## Increased contact guidance and/or cell-cell repulsion in trailing cells promotes robustness in collective migration

The emergence of stream breaks (i.e. separation between leading and trailing cells) in our simulations leads us to ask whether NCCs can recover coherent stream patterns by upregulating or downregulating certain migratory mechanisms. We address this question by simulating conditions that drive breaks and identifying parameter combinations that can rescue them. A successful rescue experiment is defined to occur when the distance between leader and follower cells is less than two NCC diameters (here, 30 μm). This threshold represents a distance slightly larger than that at which NCCs can interact through cell filopodia, here, 27.5 μm.

Of all the tested combinations of perturbations (*Figure 5A*), we find that stream breaks can be rescued by increasing the degree to which cell trajectories are influenced by contact guidance. Upregulating contact guidance in either non-secretory cells or in both cell types (by decreasing the cell-ECM weight, $\chi$, from 0.5 to 0.33) reduces distances between leader and follower cells, regardless of what other ABM parameters are altered. Thus, stream resistance to fragmentation is sensitive to the strength of the contact guidance cues NCCs receive from nearby ECM. Simulations demonstrate that enhancing contact guidance in both cell types (*Figure 5B*) creates a robust single, ribbon-like stream, with approximately constant width and density.

The only other perturbation that successfully prevents the formation of stream breaks occurs when leader cells exert weaker repulsive forces than followers. However, these streams are less robust to perturbations in other mechanisms. Furthermore, individual realizations suggest that when leaders exert weaker repulsive forces than followers (*Figure 5C*), multiple stream branches and/or single-file cell chains may form, even though the distance between cells is approximately constant. Thus, when trailing cells exert stronger repulsive forces than leading cells, additional mechanisms may be required to prevent the emergence of multiple branches and/or single-file cell chains.

## Discussion

We demonstrate here how dynamic deposition and remodeling of FN by a highly invasive cell population, such as the neural crest, can promote collective cell migration over long distances. Our simulations were motivated by several new experimental observations that: (i) leader chick cranial NCCs have enhanced expression of FN distinct from followers (*Morrison et al., 2017*), (ii) FN protein expression is punctate and disorganized in the mesoderm prior to NCC emigration, (iii) FN becomes filamentous after leader NCCs invade, and (iv) proper levels of FN within the cranial migratory pathways are required for NCC migration (*Figure 1*). These discoveries led us to develop a novel mathematical model accounting for the relationship between FN matrix remodeling and cell-cell communication. We built our model from existing theories that exploratory NCCs at the front of migrating collectives convey directional information to trailing cells. Our in silico approach allowed us to rapidly test and

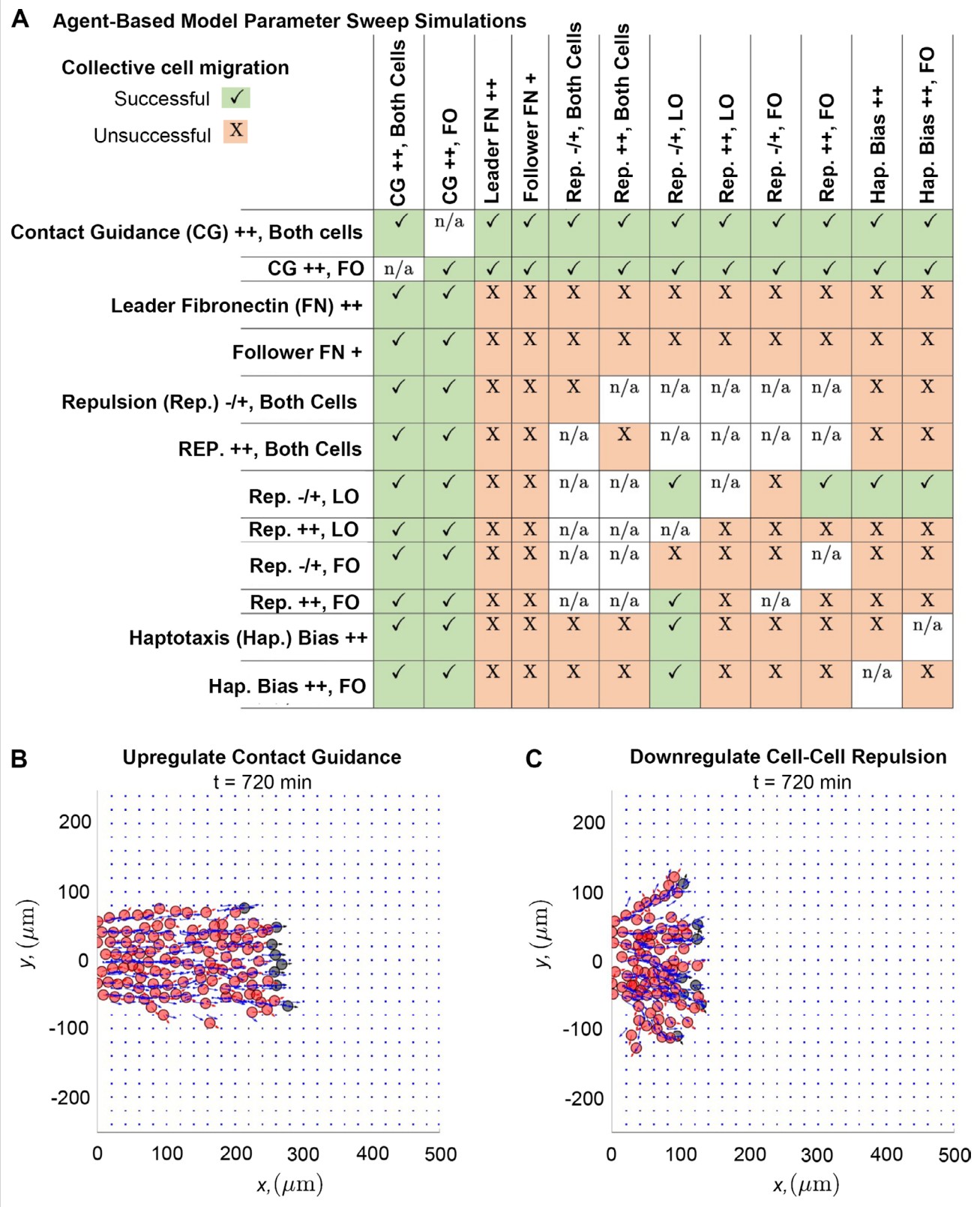

**Figure 5.** Upregulating contact guidance and/or downregulating secretory cell-cell repulsion prevents stream breakage. A summary (**A**) of experiments that either successfully (checkmarks) or unsuccessfully (**X**) prevent secretory and non-secretory cells from separating more than two cell diameters (30 μm) in at least 100 of 200 agent-based model (ABM) realizations (n/a means that the parameter combination is impossible to implement). Individual simulations suggest that cells can migrate as a single discrete stream, as when both cell types upregulate contact guidance (**B**), or as multiple single-

*Figure 5 continued on next page*

Figure 5 continued

cell chains as in the case where secretory cells downregulate cell-cell repulsion (**C**). Abbreviations in (**A**): CG ++, Both cells (both cell types upregulate contact guidance, $\chi = 0.33$); CG ++, FO (only non-secretory cells upregulate contact guidance, $\chi = 0.33$); leader FN ++ (leader secretory cells secrete FN puncta more often, $T_{ave} = 10$ min); follower FN + (trailing cells are enabled to secrete FN at a rate $T_{ave} = 90$ min); Rep. –/+, both cells (both cell types downregulate cell-cell repulsion, $c_i = 0.05$); Rep. ++, both cells (both cell types upregulate cell-cell repulsion, $c_i = 2$); Rep. –/+, LO (only leader secretory cells downregulate cell-cell repulsion, $c_i = 0.05$); Rep. ++, LO (only leader secretory cells upregulate cell-cell repulsion, $c_i = 2$); Rep. –/+, FO (only follower non-secretory cells downregulate cell-cell repulsion, $c_i = 0.05$); Rep. ++, FO (only follower non-secretory cells upregulate cell-cell repulsion, $c_i = 2$); Hap. Bias ++ (both cell types sample less noisy haptotaxis cues from the von Mises distribution, $\gamma_{FN} = 3000$); Hap. Bias ++, FO (only follower non-secretory cells sample less noisy haptotaxis cues from the von Mises distribution, $\gamma_{FN} = 3000$).

compare the effect of different mechanisms, such as haptotaxis and contact guidance, on distinct subpopulations of migrating NCCs. Using simulated 'rescue' experiments, we identified how upregulation of certain mechanisms led to re-establishment of NCC spacing and collective cell migration after changes in FN deposition and remodeling. Together, this study elucidated mechanistic aspects of how FN deposition and remodeling in an embryonic microenvironment can lead to robust collective cell migration, suggesting its critical role and a future set of integrated in vivo and in silico experiments.

In agreement with mathematical models for collective cell migration in mature ECM, we found that contact guidance and ECM remodeling are sufficient to support collective behavior (*Painter, 2009*), with the establishment of ECM fibers an important step in enabling long-distance migration (*Azimzade et al., 2019*; *Suveges et al., 2021*; *Metzcar et al., 2022*). Upregulating contact guidance increases the distances that cells travel, at the cost of increasing cell separation. Additionally, we found that cells are capable of directed migration by remodeling an initially isotropic ECM (*Park et al., 2020*). ECM remodeling also created a 'memory' for trailing cells in the ABM, since non-secretory cells were still able to migrate along the paths of leading cells even when stream breaks occurred. This is consistent with in vitro observations of individual cell migration (*d'Alessandro et al., 2021*). We further showed in our ABM that knocking out FN expression significantly reduces the number of NCCs that successfully migrate along the target corridor, which agrees with in vivo experiments of cardiac mice NCCs (*Wang and Astrof, 2016*). Furthermore, cells in the ABM traveled in the direction of greatest FN concentration, with fibers preferentially oriented in the direction of the stream migration similar to in vitro experiments (*Newgreen, 1989*; *Rovasio et al., 1983*). In future work, we plan to experimentally test the model prediction that increased production of FN by follower cells leads to stream separation.

The interactions between cells and the ECM resemble a 'long-range attraction, short-range repulsion' phenomenon found in other frameworks for NCC migration based on contact inhibition of locomotion, where cells repel each other along the direction of contact and attract each other via local chemoattractant secretion (*Carmona-Fontaine et al., 2011*; *Woods et al., 2014*; *Szabó et al., 2016*; *Szabó et al., 2019*; *Khataee et al., 2021*). Unlike those models, however, upregulating cell-cell repulsion did not always increase the distance migrated by cells. This difference likely arises from the incorporation of alignment cues from ECM remodeling in our ABM as well as the choice to stochastically sample the cell-cell repulsion force direction, as this causes cell-cell repulsion to act antagonistically to mechanisms promoting persistent migration such as contact guidance. It would be interesting to investigate whether other in silico representations of cell-cell repulsion within a dynamic fibrous ECM would have similar context-dependent effects on NCC behavior, or whether they would necessarily promote long-distance collective cell migration. We plan to address this question in future work.

Further in silico experiments demonstrated the importance of ECM remodeling by leaders of an NCC stream. NCCs migrated most persistently when leader and follower cells specialized in different roles, with cells at the front guiding NCC streams via FN remodeling. This suggests that future experiments that either knock down FN production in leaders only or disrupt an NCC's ability to align an FN fibril will disrupt collective cell migration in vivo. Simulated rescue experiments for stream breaks also suggest more robust streams may result when follower cells travel faster along leader-created paths by upregulating contact guidance and/or cell-cell repulsion. Evidence for such specialization exists in vivo, as chick cranial NCCs exhibit different morphologic and transcriptomic profiles depending on their location within the stream (*Teddy and Kulesa, 2004*; *McLennan et al., 2015a*; *Morrison et al., 2017*). Our findings do not rule out the possibility, however, that leader and follower cells may switch position and behavior as suggested by identification of Delta-Notch signaling as a potential pathway for dynamically modulating leader/follower cell types in zebrafish trunk NCCs (*Richardson*

*et al., 2016*; *Alhashem et al., 2022*). Since our focus here was to evaluate the potential importance of the ECM, and specifically FN, on NCC migration, we did not consider such phenotype switching in the ABM. However, we aim in the future to extend our modeling framework to consider cell shuffling at the front and dynamic switching between cell phenotypes, as this presents another mechanism by which robust streams may emerge.

The pre-existing FN distribution, and its possible impact on cell jamming, appeared to be a critical component in establishing collective cell migration. As might be expected from in vitro knockdown studies of the FN matrix (*Rovasio et al., 1983*), cells fail to migrate long distances within sparser matrices in the ABM. Interestingly, we found cells could overcome such jammed states by upregulating contact guidance, FN secretion, and/or cell-cell repulsion, suggesting a role for ECM remodeling to resolve jamming conditions. In future work we will quantify the pre-existing FN distribution in fluorescently labeled chick mesoderm in thin chick tissue cryosections to input a more accurate initial FN distribution and take advantage of targeted electroporation into the chick mesoderm (*McLennan and Kulesa, 2019*) in advance of NCC emigration to perturb the FN prepattern and test these model predictions.

We acknowledge that the ABM does not capture all the complexities of NCC migration and has several limitations. It relies on the assumptions that cells migrate in a 2D domain, do not divide, and only transiently contact their neighbors. While such criteria are likely to be satisfied for the motivating example of chick cranial NCC migration and for some in vitro experiments on NCCs (*Rovasio et al., 1983*), they may not be valid for other populations of NCCs such as those in the gut, where proliferation plays a significant role in collective migration (*Simpson et al., 2007*). We also incorporated a relatively small number of mechanisms into the ABM, such that we did not address the possible roles of phenotype switching and cell shuffling on NCC streams (*McLennan et al., 2012*; *McLennan et al., 2015b*; *Schumacher, 2019*). Furthermore, our implementation of cell-cell repulsion differs from those previously used in models incorporating contact inhibition of locomotion (*Colombi et al., 2020*). However, while cells in our ABM can potentially align with each other at long ranges, they repel each other at short range, with increases in cell-cell repulsion parameter values leading to greater intercellular distances. We therefore do not anticipate that other formulations of cell-cell repulsion will lead to results that are significantly different from those described here.

We plan to extend the ABM to account for, and explore the impact of, 3D migration in future work. Indeed, many equations governing cell behavior will not change when going from a 2D to a 3D environment. There will be some differences, however. For instance, the orientation of FN fibers will need to be described with 3D spherical coordinates rather than 2D polar coordinates. This will consequently affect the differential equation describing how cells update the fiber orientation (*Germann et al., 2019*). The distributions from which one samples the directions of the haptotactic and cell-cell repulsion forces will also have to change to a 3D von Mises-Fisher distribution from a 2D von Mises distribution (*Sra, 2012*). It would be interesting to evaluate the impact of 3D settings on cell migration, as the impact of jamming may be reduced in a scenario where cells have more freedom to travel out of the plane.

Despite these limitations, the ABM proved useful in providing experimentally testable predictions for in vivo NCC migration. The observation that NCC streams often exhibited excessive lateral migration led us to implement a simple 'guiding force' to ensure cells traveled along the correct target corridor. In vivo signals known to affect chick cranial NCC migration, such as chemotaxis up VEGF gradients and anisotropic domain growth (*McLennan et al., 2012*; *McKinney et al., 2020*), may play such a role in correctly steering cell streams. However, these mechanisms alone cannot account for the robustness of streams, as we found in the ABM that the guiding forces made stream breaks more likely to occur. These results are consistent with those obtained from more detailed cell-induced chemotaxis models for collective migration (*Giniūnaitė et al., 2020*). Simulated rescue experiments with our ABM, which constitute one of the first comprehensive surveys of mechanisms enabling robust collective NCC migration, suggested that leader-follower separation can be averted when cells upregulate contact guidance or when leading and trailing cells have different strengths of cell repulsion. Future studies will be needed to test these predictions in vivo, but such results demonstrate the power of data-driven mathematical modeling to explore biological hypotheses and assist with experimental design.

## Materials and methods

### Description of the mathematical model

Overview

The mathematical model depicts individual cells as agents that are spherically symmetric with a user-specified constant cell body radius, $R_{cell}$ , that determines their surface area and volume (*Figure 2—figure supplement 1A*). Such frameworks are known as 'overlapping spheres' models in the literature because the spheres can potentially intersect, with the area of overlap between two cells being proportional to the repulsive force between them. We implemented and solved the ABM using Phys-iCell (version 1.7.1), a C++ library for simulating overlapping spheres models (*Ghaffarizadeh et al., 2018*), because this software specializes in efficiently running this type of framework and has fewer dependencies compared to other libraries such as Chaste (*Mirams et al., 2013*).

Each cell in the ABM is equipped with another radius, $R_{filo} > R_{cell}$ , which represents the maximum distance that a cell can extend a filopodium from its cell center (*Figure 2—figure supplement 1A*). This is intended to represent the fact that cells have variable filopodia lengths, as NCCs can sense all FN puncta and fibers located between their cell body and filopodial radii. Puncta and fibers within a cell body radius are 'covered' by the NCC and not able to be sensed. We incorporate the filopodial radius in PhysiCell by setting the default radius of interaction to be equal to $R_{filo}$.

We consider two cell phenotypes in the ABM: 'leader' or 'secretory' cells, which are initially placed at the front of migrating collectives and can deposit new FN puncta, and 'follower' or 'non-secretory cells' which enter the domain at later time points and cannot create new FN puncta (*Figure 2—figure supplement 1*). The ability to introduce new FN puncta in the domain constitutes the only difference between the two cell types, as both may establish fibers from existing puncta (see below for more details).

We idealize NCC migration as a 2D process occurring in a plane, due to observations in the chick head that suggest the environment in which cells migrate is shallow, that is, approximately three to four cell diameters in depth (*Kulesa and Fraser, 1998*). Unlike existing frameworks for NCC migration, which use rectangular domains confining cells to a migratory corridor of up to 150 μm in width (*McLennan et al., 2012*; *Szabó et al., 2016*), cells in the ABM invade an open square domain of area 250,000 $\mu m^2$ , where the left-hand boundary represents the neural tube. Since the FN matrix prior to NCC entry does not appear to favor a particular migration direction, we model the initial FN matrix as a punctate isotropic square lattice and specify the uniform spacing, $\lambda_{FN}$ , between puncta (*Figure 2—figure supplement 1B*). We assume that FN does not diffuse or decay, as these processes occur on a longer timescale than that of NCC migration (*Kulesa et al., 2004*; *McKinney et al., 2020*). Prior research indicates that at most 10% of NCCs undergo mitosis during the first 12 hr of migration (*Ridenour et al., 2014*), which suggests cell division is a negligible contributor to collective cell migration and hence can be ignored in the mathematical model.

### Determining cell velocities in the mathematical model

In the ABM, the velocity, $\boldsymbol{v}^{(i)}$ , of cell $i$ is determined by an overdamped Newton's law. This assumes that cells can be approximated as point particles, that their movement can be determined by balancing forces acting on their centers of mass, and that inertial effects are negligible compared to forces arising from friction, cell-cell repulsion, and cell-ECM interactions. Incorporating the above assumptions leads to the following equation for the velocity:

$$\boldsymbol{v}^{(i)} \approx \frac{1}{\eta}\boldsymbol{F}^{(i)}_{ECM} + \frac{1}{\eta}\boldsymbol{F}^{(i)}_{rep}.$$

We have further assumed without loss of generality throughout this work that the coefficient of friction, $\eta$, is equal to one. We remark that the model is formulated in dimensional (not dimensionless) units. The positions of the cell centers are computed from their respective velocities using a second-order Adams-Bashforth method.

To prevent unrealistic cell velocities, we specify a maximum cell speed that depends on whether a cell senses an FN punctum or fiber. Cells that sense at least one FN punctum or fiber within their vicinity admit a larger maximum speed than cells that do not sense FN. This is intended to model a chemokinesis-like mechanism observed in vivo, where cells appear to travel faster in FN-rich areas.

In the ensuing paragraphs, we describe how the cell-ECM force for cell $i$, $\boldsymbol{F}_{ECM}^{(i)}$, and the cell-cell repulsive force, $\boldsymbol{F}_{rep}^{(i)}$, are determined in the ABM from all neighboring NCCs, FN puncta, and FN fibers sensed by the agent. A flowchart depicting how cell velocities are calculated is provided in *Figure 2—figure supplement 1C*.

## Description of how the cell-ECM force is calculated

The direction of the cell-ECM force is determined by two vectors of unit length. The first, $\hat{\boldsymbol{F}}_{hap}$, corresponds to haptotaxis and biases NCC migration toward increasing densities of FN puncta and fibers. The second, $\hat{\boldsymbol{F}}_{cg}$, represents contact guidance and aligns cell velocities along the direction of FN fibers. For simplicity, we take the direction of the resultant cell-ECM force, $\hat{\boldsymbol{F}}_{ECM}^{(i)}$, as a linear combination of these vectors weighted by a fixed parameter $0 \leq \chi \leq 1$:

$$\hat{\boldsymbol{F}}_{ECM}^{(i)} = \frac{\left( \chi \hat{\boldsymbol{F}}_{hap} + (1 - \chi) \, \hat{\boldsymbol{F}}_{cg} \right)}{\left\| \chi \hat{\boldsymbol{F}}_{hap} + (1 - \chi) \, \hat{\boldsymbol{F}}_{cg} \right\|},$$

where $\|\cdot\|$ is the Euclidean norm. The magnitude of the cell-ECM force, $F_{ECM}^{(i)}$, is determined by the number and position of neighboring FN puncta and fibers sensed by the cell. For simplicity, we have assumed that the force exerted by a single puncta or fiber decreases quadratically with increasing distance from the cell center. The magnitude of the total cell-ECM force is given by

$$F_{ECM}^{(i)} = \left\| \sum_{j=1}^{N_{FN}} \left( 1 - \frac{\left\| \boldsymbol{x}_i - \boldsymbol{p}_j \right\|}{R_{filo}} \right)^2 \frac{\left( \boldsymbol{x}_i - \boldsymbol{p}_j \right)}{\left\| \boldsymbol{x}_i - \boldsymbol{p}_j \right\|} \right\|,$$

where $\|\cdot\|$ is the Euclidean norm, $N_{FN}$ the number of FN puncta and fibers sensed within an annular region of inner and outer radii $R_{cell}$ and $R_{filo}$, respectively, $\boldsymbol{p}_j$ the position of punctum/fiber $j$, and $\boldsymbol{x}_i$ the position of cell $i$. Note that this assumes any FN puncta or fibers that are 'covered' by a cell cannot be sensed. The total cell-ECM force is given by $\boldsymbol{F}_{ECM}^{(i)} = F_{ECM}^{(i)} \hat{\boldsymbol{F}}_{ECM}^{(i)}$.

## Description of ECM remodeling in the ABM and how this determines the direction of contact guidance

The direction of contact guidance is given by the average orientation of all FN fibers sensed within an annular region of inner radius $R_{cell}$ and outer radius $R_{filo}$ of the cell:

$$\boldsymbol{F}_{cg} = \frac{1}{N_{fibers}} \sum_{j=1}^{N_{fibers}} \begin{pmatrix} \cos \left( \phi_j \right) \\ \sin \left( \phi_j \right) \end{pmatrix},$$

where $\phi_j$ represents the angle fiber $j$ makes with the $x$-axis and $N_{fibers}$ the number of fibers located within the annular region described previously. The unit contact guidance direction, $\boldsymbol{F}_{cg}$, can be obtained from the above vector by dividing by the magnitude of $\boldsymbol{F}_{cg}$. Since FN fibers do not initially exist in the model, however, we must stipulate how they are assembled from immature FN puncta by motile NCCs. We thus distinguish between two types of FN in our modeling framework: puncta and fibers. Fibers are nearly identical to FN puncta but are equipped with an additional orientation, $\phi$. Migrating cells assemble fibers from existing puncta when they pass over them (*Figure 2—figure supplement 2*). The initial orientation of the new fiber is chosen to be aligned along the direction of the cell velocity. Any NCC may further alter the orientation of existing fibers by passing over it: in this case, the cell alters the fiber orientation, $\phi$, to its current velocity direction, $\theta$, according to the ordinary differential equation

$$\frac{\mathrm{d}\phi}{\mathrm{d}t} = \frac{\ln (2)}{T_{half}} \sin \left( \theta - \phi \right),$$

where the user-defined parameter $T_{half}$ represents the approximate time it takes to halve the angle between the fiber orientation and cell velocity vectors (*Dallon et al., 1999*). A cartoon depiction of this process is shown in *Figure 2—figure supplement 2*.

Additional ECM remodeling can occur when secretory cells deposit new FN puncta. This action, which creates a new FN punctum at the cell center location, occurs according to time intervals drawn from an exponential distribution with user-specified mean $T_{ave}$. After a cell deposits an FN punctum, it samples a time interval, $t_r$, from this distribution. The cell then secretes another FN puncta when the time since its last secretion event exceeds $t_r$, provided at least one other FN molecule is located within an annular region of inner radius $R_{cell}$ and outer radius $R_{filo}$ of the cell. The latter criterion models in vivo observations of FN deposition.

## Description of how the direction of haptotaxis is determined

The direction in which the cell biases its cell-ECM force in response to haptotaxis, $\hat{F}_{hap}$, is sampled from a von Mises distribution (*Mardia and Jupp, 1999*) with parameters $\text{Arg}\left(\nabla S_{FN}\right)$ and $\|\nabla S_{FN}\|$, where

$$\nabla S_{FN} = \gamma_{FN} \left[ \sum_{j=1}^{N_{FN}} \nabla \left\{ \exp\left( -\frac{\left\| x_i - p_j \right\|^2}{2\left(R_{filo} - R_{cell}\right)^2} \right) \right\} \right].$$

This approach follows that originally introduced by *Binny et al., 2016*, to yield directional information from discrete distributions. In this case, the parameters are chosen such that cell migration is biased toward increasing densities of FN molecules but becomes more random if the cell senses FN located far from its center of mass.

## Description of how the cell-cell repulsive force is calculated

The direction of the cell-cell repulsion force, $F_{rep}^{(i)}$, is similarly sampled from a von Mises distribution with parameters $\text{Arg}\left(-\nabla S_{cell}\right)$ and $\|-\nabla S_{cell}\|$, where

$$\nabla S_{cell} = \gamma_{cell} \left[ \sum_{j=1}^{N_{cell}} \nabla \left\{ \exp\left( -\frac{\left\| x_i - x_j \right\|^2}{2\left(R_{cell}\right)^2} \right) \right\} \right],$$

and $N_{cell}$ is the number of cells within a neighborhood of radius $R_{filo}$ around cell $i$. This allows cells to bias their migration toward low-density regions, while still enabling cells to align with each other when they are further away. The magnitude of the force is determined in a similar fashion to that of the cell-ECM force, and is given by

$$F_{rep}^{(i)} = \left\| \sum_{j=1}^{N_{cell}} c_i \left( 1 - \frac{\left\| x_i - x_j \right\|}{R_{filo}} \right)^2 \frac{(x_i - x_j)}{\left\| x_i - x_j \right\|} \right\|,$$

where $c_i$ denotes the strength of cell-cell repulsion. The total cell-ECM force is given by $\boldsymbol{F}_{rep}^{(i)} = F_{rep}^{(i)} \boldsymbol{F}_{rep}^{(i)}$.

## Initial and boundary conditions of the model

We model NCC migration from the neural tube by initializing a column of non-intersecting secretory cells a distance $R_{cell}$ away from a sub-interval of the left-hand boundary (with user-defined width $l_{entr}$ centered along the line $y = 0$), as depicted in *Figure 2—figure supplement 1B*. This 'entrance strip' is intended to represent the emergence of NCCs from a discrete subset of the neural tube. To guarantee that each NCC senses at least one FN punctum at the beginning of simulations, so that migration can occur, the first column of the FN lattice is placed a distance $R_{cell} + 0.5R_{filo}$ to the right of the secretory cell centers. Non-secretory trailing cells subsequently enter the domain at the same locations as the initial secretory cells, provided space is available. For the parameter values listed in *Table 1*, seven leader cells were initialized. We found that the number of trailing cells that enter the domain using these parameter values ranged from 1 to 140, depending on the success of migration (*Figure 2—figure supplement 8*).

Cells already within the domain experience reflective conditions at all boundaries, which in practice is implemented as if they 'bounce back' from the wall. This ensures that no cell can exit the domain and is a discrete analogue of a no-flux boundary condition.

## Computational performance

The walltime for a single ABM realization varied between about 90 s and 2 min, depending on the number of cells that enter the domain and the number of FN puncta secreted. Simulations were performed on a workstation composed of four cores (Intel(R) Core(TM) i5-7500T CPU @ 2.70 GHz).

## Data analysis

Violin plots were generated with Matlab (version 2021a), using open-source methods from *Bechtold, 2016*. Violin plots are designed to yield the same information as box plots, but also use kernel density estimation to calculate the approximate distribution of the summary statistic. Statistical significance of different summary statistic distributions compared to the baseline (WT) parameter regime was assessed via a Mann-Whitney U-test using the *ranksum* function in Matlab (version 2021a). The Mann-Whitney U-test determines whether two distributions are significantly different by measuring the probability that a given sample from one distribution is less than that of the other. We have also corrected p-values for multiple comparisons by applying a conservative Bonferroni correction, in which we multiply the raw p-values by the number of statistical tests performed, to reduce the false discovery rate (*Goeman and Solari, 2014*).

## Parameter values

*Table 1* presents a list of the parameters used for the 'baseline parameter regime' discussed in the Results section (a.u.=arbitrary units). The sources of these values are listed in the table.

## Guiding force implementation in the discrete model

The 'guiding force' discussed in the Results section is oriented along the horizontal axis, such that its vector of unit magnitude is given by

**Table 1.** Parameter values corresponding to the baseline parameter regime listed in the main text.

| Parameter | Value (units) | Meaning | Reference |
|---|---|---|---|
| $R_{filo}$ | 27.5 (µm) | Filopodial radius of cells (measured from the cell center of mass) | *Colombi et al., 2020*; *McLennan and Kulesa, 2010* |
| $R_{cell}$ | 7.5 (µm) | Cell body radius | *Szabó et al., 2019*; *McLennan et al., 2020* |
| $l_{entr}$ | 120 (µm) | Length of neural tube section from which cells emerge | *Szabó et al., 2019*; *McLennan et al., 2012* |
| $T_{ave}$ | 30 (min) | Average time for cells to secrete FN | *Rozario et al., 2009*; *Davidson et al., 2008* |
| $T_{half}$ | 30 (min) | Average time it takes for cells to adjust the orientation of FN fibers | *Rozario et al., 2009*; *Davidson et al., 2008* |
| $s_{FN}^{max}$ | 0.8 (µm/min) | Maximum cell speed when cells sense FN | *Ridenour et al., 2014* |
| $s_{off}^{max}$ | 0.05 (µm/min) | Maximum cell speed when cells do not sense FN | Selected to be an order of magnitude smaller than $s_{FN}^{max}$ |
| $\Delta t$ | 0.1 (min) | Time step | Selected based on preliminary ABM simulations |
| $\gamma_{FN}$ | 300 (a.u.) | Height of Gaussian function used to parameterize the von Mises distribution for the haptotaxis force | Selected based on preliminary ABM simulations |
| $\gamma_{cell}$ | 100 (a.u.) | Height of Gaussian function used to parameterize the von Mises distribution for the cell-cell repulsion force | Selected based on preliminary ABM simulations |
| $\lambda_{FN}$ | 20 (µm) | Spacing between FN puncta of the initial lattice | Selected to ensure cells sense at least one FN punctum at all times |
| $\chi$ | 0.5 (a.u.) | Fraction by which the cell-ECM force is weighted in the direction of the haptotaxis force | Selected based on global sensitivity analysis results |
| $c_i$ | 0.5 (a.u.) | Cell-cell repulsion force strength | Selected based on global sensitivity analysis results |

$$\hat{F}_{guid} = \begin{pmatrix} 1 \\ 0 \end{pmatrix}.$$

Since we cannot deduce the magnitude of the guiding force a priori, we opt to have the guiding force only affect the direction, not the speed, of cell motion. The cell velocity is generalized such that its unit direction is given by

$$\hat{v}^{(i)} \approx \frac{(1-z)\left(\frac{1}{\eta}F_{ECM}^{(i)} + \frac{1}{\eta}F_{rep}^{(i)}\right) + z\hat{F}_{guid}}{\left\|(1-z)\left(\frac{1}{\eta}F_{ECM}^{(i)} + \frac{1}{\eta}F_{rep}^{(i)}\right) + z\hat{F}_{guid}\right\|},$$

where $0 \leq z \leq 1$ is a user-defined weight controlling the 'strength' of the guiding force, and $\|\cdot\|$ denotes the Euclidean norm. The speed of cell $i$ is given by

$$v^{(i)} = \min\left\{s_{max}, \; \left\|\frac{1}{\eta}F_{ECM}^{(i)} + \frac{1}{\eta}F_{rep}^{(i)}\right\|\right\},$$

where $s_{max}$ denotes the user-defined maximum possible cell speed. When $z = 0$, cells move without a guiding force. By contrast, when $z = 1$, cells move strictly to the right.

## Computational model pseudocode

1. Create initial distributions of FN puncta and secretory NCCs.
2. Initialize edge agents (these agents are located at the same initial position of leader cells, but are stationary, do not interact with NCCs, and are only used to determine when new cells enter the domain – see Step 5(e) below).
3. **For** all secretory cells:
   a. Sample a time interval $t_r$ from an exponential distribution, with user-specified mean $T_{ave}$.
4. Set $t = 0$.
5. **While** $t < 720$ min:
   a. **For** all cells, compute their new velocities:
      i. **For** all FN puncta and fibers that lie outside a distance $R_{cell}$ and within a distance $R_{filo}$ :
         1. Compute the gradient of the FN density, $\nabla S_{FN}$.
         2. Record that the cell senses FN and set the maximum cell speed to $s_{FN}^{max}$.
         3. Sample $\hat{\boldsymbol{F}}_{hap} \sim$ von Mises $\left(\nabla S_{FN}, \|\nabla S_{FN}\|\right)$.
         4. If there are any FN fibers, compute $\boldsymbol{F}_{cg}$ using the equation listed in the Materials and methods section and divide by its magnitude to determine the corresponding unit vector, $\hat{\boldsymbol{F}}_{cg}$. Otherwise, set $\hat{\boldsymbol{F}}_{cg}$ equal to 0.
         5. Compute the direction of the resulting cell-ECM force, $\hat{\boldsymbol{F}}_{ECM}^{(i)}$ , and its magnitude, $F_{ECM}^{(i)}$ , to obtain the total force $\boldsymbol{F}_{ECM}^{(i)} = F_{ECM}^{(i)}\hat{\boldsymbol{F}}_{ECM}^{(i)}$.
      ii. **For** all neighboring cells whose centers lie within a distance $R_{filo}$ :
         1. Compute the gradient of the cell density, $\nabla S_{cell}$.
         2. Sample $\hat{\boldsymbol{F}}_{rep} \sim$ von Mises $\left(-\nabla S_{cell}, \|-\nabla S_{cell}\|\right)$.
         3. Compute the magnitude of the resulting cell-cell repulsive force, $F_{rep}^{(i)}$ , to obtain the total force $\boldsymbol{F}_{rep}^{(i)} = F_{rep}^{(i)}\hat{\boldsymbol{F}}_{rep}^{(i)}$.
      iii. If the cell did not sense any FN puncta or fibers in step (i)(2), then set its maximum cell speed to be $s_{off}^{max}$. If it also did not sense any neighboring cells in step (ii), then do not change its current velocity and proceed to step 5(b).
      iv. If the cell sensed at least one FN punctum/fiber or at least one other cell, then compute the new cell velocity, $\boldsymbol{v}^{(i)}$.
      v. Reset the magnitude of the cell velocity to the maximum cell speed if the former quantity exceeds that value.
      vi. If the cell experiences a guiding force, adjust the direction of the velocity according to the equation listed above.
      vii. Record the new cell velocity.
   b. Apply a second-order Adams-Bashforth algorithm to update the position of all cell centers, given their velocities.
   c. Secrete new FN puncta in the domain:
      i. **For** all secretory NCCs:
         1. If the time since the cell last secreted FN exceeds $t_r$ AND the cell has sensed at least one FN punctum or fiber:
            a. Place a new FN punctum at the cell center.
            b. Sample a new time interval $t_r$ from an exponential distribution with mean $T_{ave}$.
   d. Apply reflective boundary conditions to all cells that have exited the domain.
   e. Add non-secretory cells to the domain:
      i. **For** all edge agents (see step 2):
         1. If the edge agent center is not covered by another NCC, then add a new non-secretory cell whose center coincides with that of the edge agent.
   f. Remodel the FN matrix:
      i. **For** all FN puncta and fibers that lie within a distance $R_{cell}$ of an NCC:
         1. If the FN substrate is a punctum, then convert it to a fiber by equipping it with an orientation in the direction of the current velocity of the cell.
         2. If the FN substrate is a fiber, then use Euler's method to solve the ODE for $\frac{d\phi}{dt}$ to find its new orientation.
   g. Set $t = t + \Delta t$.
   **End while** loop

## Sensitivity analysis

Pearson and partial Spearman rank correlation coefficients were computed using the *corrcoef* and *partialcorr* functions in Matlab version 2022a, respectively. Sobol indices were calculated with eFAST, which exploits periodic sampling strategies and fast Fourier transforms to accelerate the computation of these metrics (*Saltelli et al., 1999*). The eFAST algorithm calculates Sobol indices in the following manner: first, the algorithm samples parameter values from periodic distributions that ensure the values are approximately uniform within user-defined ranges; the frequencies of such periodic distributions

**Table 2.** Parameter values and ranges used to generate the extended Fourier amplitude sensitivity testing (eFAST) results.

| Parameter | Range of values considered in eFAST | Units |
| --- | --- | --- |
| $R_{filo}$ | [30, 75] | μm |
| $\chi$ | [0.25, 1] | Arbitrary units |
| $c_i$ | [0, 5] | Arbitrary units |

(and their higher-order harmonics) are unique for each parameter. After computing model outputs and summary statistics for each parameter regime, the algorithm next performs a Fourier transform on the variance of each statistic. The algorithm finally calculates Sobol indices by measuring the fraction of the variance that is generated by the Fourier coefficients uniquely associated to each parameter.

We examined the sensitivities of $R_{filo}$, the maximum filopodial length, $\chi$, the ABM weight controlling the extent to which NCC responses to the ECM are dominated by haptotaxis or contact guidance, and $c_i$, the cell-cell repulsion force strength. *Table 2* presents a list of the ranges of each parameter considered with eFAST. The range for the filopodial radius was chosen based on in vivo measurements of NCCs (*Colombi et al., 2020*; *McLennan et al., 2012*). Ranges for the cell-ECM weight, $\chi$, and the cell-cell repulsive force strength, $c_i$, were selected based on observations of preliminary ABM simulations, as the qualitative behavior of streams did not seem to be affected much when $\chi < 0.25$ and $c_i > 5$. All other ABM parameter values are fixed at the values stated in *Table 1*, with the exceptions of $R_{cell}$ (10 μm), $l_{entr}$ (150 μm), $T_{half}$ (60 min), $s_{FN}^{max}$ (0.75 μm/min), and $\lambda_{FN}$ (35 μm). We fixed these latter parameter values because their PRCCs (*Figure 2—figure supplement 9*) suggest relatively weak correlations with the maximum distance that cells travel compared to the parameters $\chi$ and $R_{filo}$. We therefore do not expect that fixing these parameter values will greatly affect the conclusions obtained from the eFAST analysis.

We perform parameter sampling and eFAST analysis using the SALib library in Python (version 3.9.1), which provides unit-tested and open-source functions for global sensitivity analysis (*Herman and Usher, 2017*). One-hundred and thirty-one values for all three parameters (in addition to a dummy variable) were sampled from uniform distributions, with summary statistics averaged across 20 ABM replicates to reduce noise. The algorithm was repeated another two times to ensure statistical significance of the Sobol indices. Statistical significance was assessed for each parameter of interest by comparing its Sobol indices with those of the dummy variable via a two-sample t-test (*Marino et al., 2008*). p-Values for PRCCs were computed via a two-tailed t-test against the null hypothesis that the PRCC is equal to zero.

## Embryos

Fertilized White Leghorn chicken eggs (Centurion Poultry, Inc, Lexington, GA, USA) were incubated in a humidified incubator at 37°C to the desired developmental stage, measured by counting somite pairs in reference to HH stage (*Hamburger and Hamilation, 1951*).

## Immunohistochemistry

Embryos were incubated in a humidified incubator at 37°C until the appropriate stage of development, then harvested and fixed overnight with 4% paraformaldehyde in PBS. After removal of extraneous tissue, embryos were prepared for whole-mount IHC by bisecting the head down the midline. Immunohistochemistry was performed by blocking in 4% bovine serum albumen and 0.01% TritonX for 2 hr at room temperature. Primary antibodies B3/D6 (Developmental Studies Hybridoma Bank) and HNK-1 (1:25, TIB-200, ATCC) overnight at 4°C in block as described in *McLennan and Kulesa, 2007*. Secondary antibody used for HNK1 was goat anti-mouse IgM (heavy stream) Alexa Fluor 488 (A-21042, Thermo Fisher Scientific) at 1:500. The samples were washed with PBS and then imaged.

## FN expression analysis

Transverse 15 μm cryosections through the r4 axial level of chick embryos stage 9 through 12 were stained with HNK1 and FN antibodies as described above. Collection of 3D confocal z-stacks (Zeiss

LSM 800) and maximum z-projection of images were performed. We measured the total distance migrated by NCCs in control and morpholino-injected embryos (*Figure 1E*). We calculated the percentage of area covered using the 'Surfaces' function of Imaris (Bitplane) to create a surface mask by manually drawing the outline of the entire second branchial arch (ba2). We then calculated the area of the HNK1 fluorescence signal using the masked ba2 surface. We set a consistent intensity threshold to the same value for each dataset, a surface grain size of 1 μm was set, the diameter of the largest sphere was set to 1 μm, and the automatic 'Surfaces' function was applied. When the percentage of area was calculated using a signal other than HNK1, we manually measured the area of the ba2 and separately manually measured the area surrounding the signal within the ba2, creating a 'Surfaces'. We calculated the percentage of the total arch the HNK-1 signal covered by comparing the two values. The percentage of distance the cells migrated was calculated by measuring the total distance to the ba2 and measured the distance migrated. The box plots in each figure (*Figure 1E and F*) were generated by using the values from each dataset indicated. The box plots and whiskers indicate the quartiles and range, respectively, of each dataset. p-Values were calculated using a standard Student's t-test.

## Gain- and loss-of-function experiments

Loss-of-function of FN was accomplished by morpholino, introduced into premigratory cranial NCCs at HH10 (*n*=15 embryos/method), eggs re-incubated for 12 hr, and intact embryos subsequently harvested for 3D confocal static imaging. For gain-of-function experiments, soluble FN was microinjected into the mesoderm adjacent to r4 at HH9 prior to NCC emigration. Eggs were re-incubated until HH13, harvested and mounted for 3D confocal static imaging. The non-injected sides of embryos were used for control comparisons. We include control morpholino images (*Figure 1—figure supplement 1*).

## Acknowledgements

WDM was supported with funding from the Keasbey Memorial Foundation (USA), the Oxford Mathematical Institute, and the Advanced Grant Nonlocal-CPD (Nonlocal PDEs for Complex Particle Dynamics: Phase Transitions, Patterns and Synchronizations) of the European Research Council Executive Agency (ERC) under the European Union's Horizon 2020 research and innovation program (grant agreement No. 883363). LAD is funded by the Eunice Kennedy Shriver Institute of Child Health and Human Development at the NIH (R37 HD044750 and R21 HD106629).

## Additional information

### Funding

| Funder | Grant reference number | Author |
| --- | --- | --- |
| European Research Council | 883363 | William Duncan Martinson |
| University of Oxford | | William Duncan Martinson |
| Keasbey Memorial Foundation | | William Duncan Martinson |
| Eunice Kennedy Shriver National Institute of Child Health and Human Development | R37 HD044750 | Lance A Davidson |
| Eunice Kennedy Shriver National Institute of Child Health & Human Development | R21 HD106629 | Lance A Davidson |

The funders had no role in study design, data collection and interpretation, or the decision to submit the work for publication.

## Author contributions
William Duncan Martinson, Conceptualization, Data curation, Software, Formal analysis, Visualization, Methodology, Writing – original draft, Writing – review and editing; Rebecca McLennan, Conceptualization, Data curation, Methodology; Jessica M Teddy, Mary C McKinney, Data curation, Methodology; Lance A Davidson, Conceptualization, Methodology, Writing – review and editing; Ruth E Baker, Conceptualization, Writing – review and editing; Helen M Byrne, Philip K Maini, Conceptualization, Supervision, Writing – review and editing; Paul M Kulesa, Conceptualization, Methodology, Writing – original draft, Writing – review and editing

## Author ORCIDs
William Duncan Martinson (ID) http://orcid.org/0000-0002-3590-606X
Lance A Davidson (ID) http://orcid.org/0000-0002-2956-0437
Ruth E Baker (ID) http://orcid.org/0000-0002-6304-9333
Paul M Kulesa (ID) http://orcid.org/0000-0001-6354-9904
Philip K Maini (ID) http://orcid.org/0000-0002-0146-9164

## Ethics
Our work did involve animal experimentation subject to ethical guidelines and according to the approved protocol IBC-2003-23-pmk.All of our chick embryology experiments were performed during the first week of development (<7 days; Hamburger and Hamilton (1951) Stages 11-16). This is significantly shorter in time than 2/3 (14days) to normal hatching (21 days). Thus, we do not describe a measure of euthanasia to avoid pain. In order to prevent incubated eggs from developing to hatching, all batches of eggs are labeled by the user with the date of the start of incubation and all incubators are checked weekly to remove eggs older than 7 days. Should the situation arise whereby an egg hatches, end of life is carried out by immediate decapitation.

## Decision letter and Author response
Decision letter https://doi.org/10.7554/eLife.83792.sa1
Author response https://doi.org/10.7554/eLife.83792.sa2

# Additional files

## Supplementary files
• MDAR checklist

## Data availability
All data from the mathematical model have been deposited in Dryad (https://doi.org/10.5061/dryad.69p8cz958). Software used for the mathematical model is available on Github at the following link: https://github.com/wdmartinson/Neural_Crest_Project (copy archived at *Martinson, 2022*).

The following dataset was generated:

| Author(s) | Year | Dataset title | Dataset URL | Database and Identifier |
|---|---|---|---|---|
| Martinson WD | 2023 | Dynamic fibronectin assembly and remodeling by leader neural crest cells prevents jamming in collective cell migration - Mathematical Model Results | https://dx.doi.org/10.5061/dryad.69p8cz958 | Dryad Digital Repository, 10.5061/dryad.69p8cz958 |

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
