## [Editor Report]

This important study presents predictions from a computational model demonstrating the impact of the extracellular matrix on collective cell migration in the neural crest. The evidence supporting the claims of the authors is solid, and the study is interesting to cell biologists exploring cell migration in different contexts.

---

## [Decision Letter]

**Decision letter after peer review:**

Thank you for submitting your article "Dynamic fibronectin assembly and remodeling by leader neural crest cells prevents jamming in collective cell migration" for consideration by *eLife*. Your article has been reviewed by 3 peer reviewers, one of whom is a member of our Board of Reviewing Editors, and the evaluation has been overseen by Jonathan Cooper as the Senior Editor.

Essential revisions:

1) Expand the model to include the ability of cells to switch phenotypes.

2) Include contact-induced repulsion.

*Reviewer #1 (Recommendations for the authors):*

The goal of this work is to investigate the mechanisms that mediate the collective migration of neural crest cells (NCCs). The authors first perform experimental studies to visualize and quantify NCC migration in chick embryos. Through knockdown or increased density of the extracellular matrix component fibronectin, the authors confirm the importance of this protein in promoting proper migration of NCCs. These data motivate the development of an agent-based model of NCC migration to quantitatively study factors that influence migration, including interactions between leader and follower cells, contact guidance, haptotaxis, secretion of fibronectin, and distribution of fibronectin in the extracellular matrix.

The experimental data are clearly presented and provide a strong basis for model development. Modeling and experiments are complementary, as the model enables a more in-depth analysis of the mechanisms driving NCC migration. The model itself is described well, including assumptions and parameters. Many modeling results are presented; though some data are left out, presumably to keep the paper focused. Some consideration of additional simulation results that can be presented to further support the conclusions is warranted.

The authors achieved the goal of delineating specific mechanisms that influence proper NCC migration. Impact on the field is likely modest; however, such predictive, mechanistic analyses are important to drive future experimental studies.

This is a well-written and interesting study. I am excited to see how experimental data motivates the development of a mechanistic ABM to explore mechanisms of cell migration. Both the data and model are clearly presented, and I applaud the authors for managing this, as it is no small feat.

I suggest some edits to refine and improve the paper.

1. Figure 2 and associated text: it is not clear how representative the two ABM realizations are. First, how many simulations were done here? Second, how many times does each realization occur? This is important to describe to provide some context for the simulation results.

2. In general, "data not shown" is present several times. This does not give the sense that the conclusions are justified. And in my opinion, some of the omitted data is critical to the study. For example, the orientation of the ECM fibers and statistics for the orientation of cells. The orientation of ECM fibers and cells are important aspects of migration; thus, results related to these points should be presented.

3. Leaving experimental testing of some model predictions somewhat reduces the impact of this paper. While it is not necessary to validate every model simulation, a careful examination of the most prominent conclusions would reveal the results that are important to test.

4. A description of the model limitations is missing. It is essential to state the limitations of the model to provide context for the readers and acknowledge the ways in which model assumptions may affect the results.

*Reviewer #2 (Recommendations for the authors):*

This theoretical work explores, by developing an agent-based model, the idea that remodelling of the extracellular matrix plays a role in collective cell migration.

Strengths:

Well-written theoretical paper that deals with the important problem of how groups of cells migrate collectively in a directional manner. The main hypothesis that extracellular matrix remodelling could play a role, not only in cell motility but, in collective cell migration is novel and the results are important, as it allows us to make predictions that could be eventually tested with future experiments. In addition, this theoretical paper is based on the migration of neural crest cells, a highly migratory cell population whose migratory behaviour has been likened to cancer invasion during metastasis. Thus, the conclusion of this paper could have implications for understanding different cells that migrate collectively, such as during embryo development, cancer invasion, or wound healing, to name a few.

Weakness:

As this is a theoretical paper, I would have considered it desirable that the authors move away from the restrictions of developing their model on a particular animal model, as they could be free of incorporating all the cellular principles discovered in a multitude of different animal models that have analysed neural crest migration (chick, zebrafish, *Xenopus*, etc). Unfortunately, the biological assumptions of the model are based only on the migration of chick neural crest, with particular emphasis on the biological findings of the experimentalist collaborator of this paper. For example, although the authors consider a cell-cell repulsion behaviour in their model, they claim that "chick cranial NCCs do not typically repel each other upon contact (Kulesa et al., 1998; Kulesa et al., 210 2004)", avoiding this behaviour in their simulations. However, a more recent paper has shown a typical cell-cell repulsion behaviour in chick neural crest (see: Li et al. (2019). in vivo Quantitative Imaging Provides Insights into Trunk Neural Crest Migration. Cell Rep. 26, 1489-1500), and this behaviour is widely documented in other species. So, instead of restricting their model to the questionable evidence that neural crest cells do not repel each other, why not make a more general model where cell repulsion could be another parameter to be explored?

An important aspect of the model is based on the clear distinction between leaders and trailing cells during neural crest collective migration. Again, this assumption in the model seems to ignore two excellent papers that unequivocally show that there are no fixed or distinct leader and trailing neural crest cells (Richardson et al., (2016). Cell Rep. 15, 2076-88; Alhashem et al. (2022). *eLife*. 11:e73550), which has been shown in chicken cephalic neural crest (Richardson et al., (2016). Cell Rep. 15, 2076-88) as well as in all other species in which high resolution of migrating neural crest has been performed. In all these species the leader and trailing cells are defined by their position within the cluster, and not by a predefined cellular state. So, leaders become leaders as soon as they reach the front of the cluster, while they become trailers when they lose this position. This more realistic dynamic behaviour of mesenchymal cells could be incorporated into their model, making it more general and not only restricted to situations in which leader and trailing cells are predefined. For example, they could make that each time a cell reaches the front position they start secreting fibronectin, and they stop it when they lose this position.

– Make the model more general, so that its assumptions are not based only on one animal model (or on the experiments of a particular group)

*Reviewer #3 (Recommendations for the authors):*

The authors use an agent-based (biological cells are modeled as computational agents) approach to explore the observed phenomenon of neural crest cell migration (NCC) in embryonic development, which is poorly understood mechanistically. Developing and implementing a 2 phenotype model off NCC population interacting with a remodelable extracellular matrix, they recapitulate many observed behaviors (specifically collective streaming of cells), finding, with compelling support, several key factors (contact guidance, etc) needed (and likewise not needed) to produce collective migration. Their modeling effort was greatly aided by in vivo experiments, detailed sensitivity analysis, and in silico experimental manipulation.

The authors' conclusions are well supported by their computational experiments as well as in vivo data and experiments. This is a well-communicated work that adds to the literature on modeling collective cell migration as well as introduces a new way to model ECM in an agent-based framework.

Strengths of this work include:

– Cross-disciplinary collaboration between computational and bench scientists.

– Selection of appropriate techniques to model phenomena of interest.

– Use of computational techniques to simulate knockdown, upregulation, and synthetic rescue of phenomena under study much of which also ties to either novel or previously published in vivo findings.

– Appropriate use of sensitivity analysis in stochastic simulations to support conclusions and guide computational experiments.

– Well-commented open-sourced software, enabling stochastic reproducibility of results as well as distribution of knowledge to the community.

– Publication of data in well-organized download, enabling community review of evidence.

The work cannot include everything and as such, the following items are not addressed in this work but may be relevant:

– As stated by the authors, there are several possibilities not excluded or explored by the presented simulations, including behavioral switching of cells or other possible methods of communication (such as a diffusing substrate).

– Likewise, as stated by the authors, not all findings supported by the modeling and simulations are currently supported by bench-side findings, leaving open the possibility that they may not be observed in in vivo studies.

– While addressed by the authors, citing the evidence that the planes of travel of these cells are narrow, 3-D simulations were not conducted. Noting that the authors discussed the jamming of cells and its impacts on invasion, developing 3-D simulations may yield different results in the context of cell jamming. It is something that could be explored, noting that having not included 3-D simulations does not take away from the conclusions in this work.

Lines 104-115 – The authors could consider (I truly mean consider) adding a late-breaking pre-print that is related to the authors' area of study to their already excellent literature review – https://www.biorxiv.org/content/10.1101/2022.11.21.514608v1. I mention it only because it is late breaking, attempting to enable the authors to include should they wish to.

Line 150 – The figure "title" does not match the majority of the area dedicated to the figure. Perhaps there could be two figures? Or a different title? "in vivo experiments and results"?

Line 152 – Developmental stage in the caption doesn't match the stage identified in Line 141. Should it?

Figure 2 and Caption (Lines 192-200)

Figure 2a and b: I found it difficult to distinguish between velocity vectors and fiber orientation vectors. Could a different color or symbol be considered to more easily distinguish them? Also, I am not seeing the additional puncta laid down by secretory cells. I have assumed that all puncta in the simulation are being visualized in the simulation stills but see only the original grid. Finally, I assume that the extra column of puncta on the right-hand side is just an artifact of visualization or something, but it may be appropriate to address why it is there. Of course, regardless it won't affect the simulations as the cells don't make it that far.

Caption – There appears to be a statistical test indicated in Figure 2C, but the caption does not identify it. The abbreviation "VE" appears in 2D but is not defined in the caption. I think it may not be defined in the manuscript at all, but of course, I may have missed it.

Figure 3 and Caption

VE isn't defined. If it's defined before, this should be fine – I point it out only in case for whatever reason the abbreviation changes in Figure 2, leaving Figure 3 as possibly the first time it's used.

Lines 469-473 – It's not clear which of these new experimental observations are from this work and which are from previous works. I may have misunderstood, but I think they are not all new in this work. Would the authors consider citing the other works here (no doubt there were previously cited but the reader may not recall all of this), enabling the reader to know which was added to this work? Or use some other indicator to identify their most recent findings being presented as a novel in this work?

Line 570 – I believe there may be a typo – I think the computational domain has an area of 250,000 um^2 (500 by 500).

Line 574 – If FN diffused or decayed at a shorter timescale than NCC migration, I feel like those phenomena would have to be accounted for. I may be misreading, but would it be that the FN processes are on a "longer" time scale than NCC migration, thus the FN can be considered unchanged during the 720 minutes of simulated time?

General comment on overview – I think it would be reasonable to highlight here how many agents become part of a simulation typically as well as the amount of simulated time to provide readers with an idea of those scales of the simulations. On a similar note, it would be reasonable to note a typical walltime as well as roughly what hardware was used to run the simulations, giving the community a chance to understand how long it might take to run similar simulations.

Table 1 – R_cell parameter – I didn't see a McLennan et al. 2020 in the citations. I may have missed it.

Reviewing the code, I see that there seem to be puncta agents in addition to the fibronectin field. How are the puncta agents used in the simulations? How do they interact with the non-diffusing substrate field? How, if at all, do their properties (their non-coding/PhysiCell-related properties) differ from the original puncta field?

In PhysiCell, there is a default value for the radius of interaction that determines where an agent looks for its neighbors. The default value is 30 um. Roughly, how did the authors address this as they did interaction testing over the longer than 30-um distances for interaction testing that would have occurred during the sensitivity analysis?

In the equations for the gradients (Del S_FN and Del S_cell), I see the motivation – cell center i is compared to all the puncta and cell centers in its vicinity. However, when viewing the expression under the summation, it seems like one would eventually calculate one single scalar, versus a field of scalars over which a gradient could be calculated. I certainly assume that I am not seeing the obvious, but could the authors please provide this reviewer with additional explanation for how or what the gradient is calculated over?

---

## [Author Response]

Essential revisions:1) Expand the model to include the ability of cells to switch phenotypes.2) Include contact-induced repulsion.Reviewer #1 (Recommendations for the authors):I suggest some edits to refine and improve the paper.1. Figure 2 and associated text: it is not clear how representative the two ABM realizations are. First, how many simulations were done here? Second, how many times does each realization occur? This is important to describe to provide some context for the simulation results.

We have clarified in the caption of Figure 2 that the two realizations shown in panels A and B are generated from the same parameter regime (albeit with different random seeds) and demonstrate the range of possible behaviors in the model. We have also added an explanation that the data points in the scatter plot of panel D were each computed from the average of N=20 ABM realizations.

2. In general, "data not shown" is present several times. This does not give the sense that the conclusions are justified. And in my opinion, some of the omitted data is critical to the study. For example, the orientation of the ECM fibers and statistics for the orientation of cells. The orientation of ECM fibers and cells are important aspects of migration; thus, results related to these points should be presented.

We do not mention the biological measurements of ECM fiber or neural crest cell orientation from the fibronectin staining patterns in transverse chick head slices. These measurements are in progress and will be included in a future publication.

We have also attached new supplementary figures and movies in our resubmission that address the in silico experiments omitted in the original text (as Reviewer #3 has noted, these omissions were done to keep to the word limit for the initial submission). Specifically, for the experiments presented in panels A and B of Figure 2, we have attached polar histograms of the fibronectin fiber distributions and the follower cell velocity orientations at the final simulation time point t = 720 min (Figure 2 —figure supplement 3). We have included videos and figures of only the fibronectin distribution for these realizations (Figure 2 —figure supplement 3, Figure 2 – videos 3 and 4). We have included scatter plots, Sobol indices plots, and PRCC bar charts showing how the average FN orientation, the nearest neighbor distance, and the range of stream widths along the y-axis change for the different parameter regimes generated with eFAST (Figure 2 —figure supplements 4-6). We have also included a figure showing the negative correlation between the nearest-neighbor distance and the distance traveled in the horizontal direction (Figure 2 —figure supplement 7), and a histogram showing the number of trailing cells that enter the domain (Figure 2 —figure supplement 8).

We have also included supplementary violin plots that show how the range of cells along the y-axis and the average nearest neighbor distance change in the experiments presented in Figure 3 of the main text (Figure 3 —figure supplements 1-3). Additionally, there are supplementary figures showing how in silico experiments in which cell-cell repulsion is perturbed from the baseline parameter regime change the distance in the x-direction and the nearest neighbor distance statistics (Figure 3 —figure supplement 4). We also show PRCCs for a larger set of ABM parameter values than those listed in the main text (Figure 2 —figure supplement 9). The PRCCs correspond to the maximum distance cells travel, the nearest neighbor distance, and the range of cells along the vertical axis.

3. Leaving experimental testing of some model predictions somewhat reduces the impact of this paper. While it is not necessary to validate every model simulation, a careful examination of the most prominent conclusions would reveal the results that are important to test.

We agree with the Reviewer and have updated the Discussion section to highlight proposed future in vivo and in vitro experiments based on the model simulations.

4. A description of the model limitations is missing. It is essential to state the limitations of the model to provide context for the readers and acknowledge the ways in which model assumptions may affect the results.

We agree with the reviewer and have added a new paragraph in the Discussion section that makes clear the assumptions and limitations of the mathematical model (e.g., its lack of phenotype switching, its relatively simplistic approach to representing fibronectin remodeling and chemotaxis, 2-D-like migration, etc.).

Reviewer #2 (Recommendations for the authors):This theoretical work explores, by developing an agent-based model, the idea that remodelling of the extracellular matrix plays a role in collective cell migration.Strengths:Well-written theoretical paper that deals with the important problem of how groups of cells migrate collectively in a directional manner. The main hypothesis that extracellular matrix remodelling could play a role, not only in cell motility but, in collective cell migration is novel and the results are important, as it allows us to make predictions that could be eventually tested with future experiments. In addition, this theoretical paper is based on the migration of neural crest cells, a highly migratory cell population whose migratory behaviour has been likened to cancer invasion during metastasis. Thus, the conclusion of this paper could have implications for understanding different cells that migrate collectively, such as during embryo development, cancer invasion, or wound healing, to name a few.Weakness:As this is a theoretical paper, I would have considered it desirable that the authors move away from the restrictions of developing their model on a particular animal model, as they could be free of incorporating all the cellular principles discovered in a multitude of different animal models that have analysed neural crest migration (chick, zebrafish, *Xenopus*, etc).– Make the model more general, so that its assumptions are not based only on one animal model (or on the experiments of a particular group).

The role of FN in other neural crest populations is not well characterized. We focused on chick cranial neural crest cells in this manuscript due to our previously undescribed findings that (i) the fibronectin distribution is punctate in the microenvironment prior to neural crest cell invasion and (ii) fibronectin is highly expressed in leader cranial neural crest cells. If the microenvironments of other neural crest populations exhibit similar traits, then we anticipate that the insights provided by our mathematical model will be applicable to them as well. However, in the absence of current data we cannot at this time extend our conclusions to more general populations of neural crest cells. We note, however, that our current framework has provided similar conclusions to in vitro models that previously investigated neural crest interactions with fibronectin (Rovasio et al., 1983). The leader-follower like behavior exhibited by the chick cranial neural crest population may also be present in other systems: to date, time-lapse analyses of NCC migratory behaviors in a number of embryo model organisms have revealed the unexpected finding that leader NCCs in discrete migratory streams in the head (Teddy and Kulesa, 2004; Genuth et al., 2018), gut (Druckenbrod and Epstein, 2007; Young et al., 2014), and trunk (Kasemeier-Kulesa et al., 2005; Li et al., 2019) are highly exploratory – extending thin filopodial protrusions in many directions, then selecting a preferred direction to move in an invasive, directed manner; follower cranial neural crest cells extend protrusions to contact leaders and other cells to maintain cell neighbor relationships and move collectively (Kulesa et al., 2008; Ridenour et al., 2014; Richardson et al., 2016). The conclusions that were obtained from the mathematical model may therefore be applicable to these more general populations, so long as the assumptions underlying the model are satisfied. We have edited the text of the Model overview section and the Results section to make clear that while the assumptions are motivated by our experimental observations of chick cranial neural crest cells, they can also apply to more general NCC populations so long as the assumptions of 2-D-like migration, punctate ECM, and non-compact cell streams are satisfied.

Unfortunately, the biological assumptions of the model are based only on the migration of chick neural crest, with particular emphasis on the biological findings of the experimentalist collaborator of this paper. For example, although the authors consider a cell-cell repulsion behaviour in their model, they claim that "chick cranial NCCs do not typically repel each other upon contact (Kulesa et al., 1998; Kulesa et al., 210 2004)", avoiding this behaviour in their simulations. However, a more recent paper has shown a typical cell-cell repulsion behaviour in chick neural crest (see: Li et al. (2019). in vivo Quantitative Imaging Provides Insights into Trunk Neural Crest Migration. Cell Rep. 26, 1489-1500), and this behaviour is widely documented in other species. So, instead of restricting their model to the questionable evidence that neural crest cells do not repel each other, why not make a more general model where cell repulsion could be another parameter to be explored?

In our extensive experience with in vivo confocal time-lapse imaging and published results of both chick cranial and trunk neural crest cell migration we have observed short- and long-range cell communication via lamellipodial and filopodial contact that stimulates cells to follow each other. We believe our peer-reviewed published results that describe these observations are not questionable evidence but represent an extensive analysis of cell behaviors (Kulesa and Fraser, 1998; Kulesa et al., 2000; Teddy and Kulesa, 2004; Kasemeier et al., 2006; Kulesa et al., 2008; Ridenour et al., 2014; McLennan et al., 2020).

We believe there may be some confusion as to how cell-cell repulsion is implemented in the model. If a cell senses a neighbor at a relatively far distance from its center of mass, then the direction of the repulsive force is sampled from a nearly uniform distribution. This reflects our desire to have cells migrate in potentially parallel directions when sensing each other at long range. However, when cells are so close as to be nearly overlapping, the distribution is much more biased, such that it is more likely that cells repel along the direction of contact. This latter feature resembles more traditional Newtonian-like approaches for representing cell-cell repulsion (Colombi et al., 2020). We have confirmed that increasing the parameter ci, which is related to the magnitude of the cell-cell repulsion force in our model, causes the nearest neighbor distance to increase (Figure 2 —figure supplement 5; Figure 3 —figure supplement 4), demonstrating that these forces correspond to cell-cell repulsion at short range.

We acknowledge that other approaches for cell-cell repulsion could be encoded in our modeling framework. However, given that increasing the cell-cell repulsion parameter, ci, increases the distances between cells, we do not anticipate that different methods for incorporating cell-cell repulsion will have a major effect on our main results concerning the importance of the fibronectin matrix. Indeed, given that chick cranial neural crest cells are not as compact as other populations (Teddy and Kulesa, 2004), it is unclear whether repulsion will play a significant role. We agree that it would be interesting to computationally evaluate different mathematical representations of cell-cell repulsion, but this is outside the scope of the current manuscript.

We have updated the Methods section and the Results section to clarify how cell-cell repulsion works in our model at long and short ranges. We have also added supplementary figures (Figure 2 —figure supplement 5; Figure 3 —figure supplement 4) that confirm that increasing parameters related to cell-cell repulsion in our model increases interparticle distances. We have updated the Discussion section to cite other approaches for mathematically incorporating cell-cell repulsion and describe our future plans to compare them.

An important aspect of the model is based on the clear distinction between leaders and trailing cells during neural crest collective migration. Again, this assumption in the model seems to ignore two excellent papers that unequivocally show that there are no fixed or distinct leader and trailing neural crest cells (Richardson et al., (2016). Cell Rep. 15, 2076-88; Alhashem et al. (2022). eLife. 11:e73550), which has been shown in chicken cephalic neural crest (Richardson et al., (2016). Cell Rep. 15, 2076-88) as well as in all other species in which high resolution of migrating neural crest has been performed. In all these species the leader and trailing cells are defined by their position within the cluster, and not by a predefined cellular state. So, leaders become leaders as soon as they reach the front of the cluster, while they become trailers when they lose this position. This more realistic dynamic behaviour of mesenchymal cells could be incorporated into their model, making it more general and not only restricted to situations in which leader and trailing cells are predefined. For example, they could make that each time a cell reaches the front position they start secreting fibronectin, and they stop it when they lose this position.

We agree with the reviewer that a current limitation of our model is its lack of dynamic switching between leader and follower types. However, the role of such phenotype switching was not our focus in the present manuscript. We aim to evaluate the impact of such phenotype switching on our ABM in future work, where we would have more space to give it the development it deserves. Based on previous implementations of phenotype switching in mathematical models for neural crest migration with cell-induced chemotactic gradients (McLennan et al., 2012; Schumacher 2019), we expect that adding such mechanisms will increase the distance that cells travel. However, this increase may be very small, given the fact that leader and follower cells in the ABM are already very similar to each other (with the only difference being that leaders can secrete new fibronectin in the domain). We have added a paragraph in the Discussion section that highlights the model limitations discussed by this reviewer and have indicated our plans for future investigations into phenotype switching.

Reviewer #3 (Recommendations for the authors):The authors use an agent-based (biological cells are modeled as computational agents) approach to explore the observed phenomenon of neural crest cell migration (NCC) in embryonic development, which is poorly understood mechanistically. Developing and implementing a 2 phenotype model off NCC population interacting with a remodelable extracellular matrix, they recapitulate many observed behaviors (specifically collective streaming of cells), finding, with compelling support, several key factors (contact guidance, etc) needed (and likewise not needed) to produce collective migration. Their modeling effort was greatly aided by in vivo experiments, detailed sensitivity analysis, and in silico experimental manipulation.The authors' conclusions are well supported by their computational experiments as well as in vivo data and experiments. This is a well-communicated work that adds to the literature on modeling collective cell migration as well as introduces a new way to model ECM in an agent-based framework.Strengths of this work include:– Cross-disciplinary collaboration between computational and bench scientists.– Selection of appropriate techniques to model phenomena of interest.– Use of computational techniques to simulate knockdown, upregulation, and synthetic rescue of phenomena under study much of which also ties to either novel or previously published in vivo findings.– Appropriate use of sensitivity analysis in stochastic simulations to support conclusions and guide computational experiments.– Well-commented open-sourced software, enabling stochastic reproducibility of results as well as distribution of knowledge to the community.– Publication of data in well-organized download, enabling community review of evidence.The work cannot include everything and as such, the following items are not addressed in this work but may be relevant:– As stated by the authors, there are several possibilities not excluded or explored by the presented simulations, including behavioral switching of cells or other possible methods of communication (such as a diffusing substrate).

Please see the response to the comments of Reviewer #2, Points 2-3.

– Likewise, as stated by the authors, not all findings supported by the modeling and simulations are currently supported by bench-side findings, leaving open the possibility that they may not be observed in in vivo studies.

Please see the response to the comments of Reviewer #1, Point 3.

– While addressed by the authors, citing the evidence that the planes of travel of these cells are narrow, 3-D simulations were not conducted. Noting that the authors discussed the jamming of cells and its impacts on invasion, developing 3-D simulations may yield different results in the context of cell jamming. It is something that could be explored, noting that having not included 3-D simulations does not take away from the conclusions in this work.

We agree with the reviewer that exploring the impact of 3-D environments would be an interesting avenue for future work. However, since assessing 3-D migration was not the primary focus of this manuscript, and because the experimental evidence that inspired the model involved narrow, 2-D-like domains, we did not consider such environments. As the reviewer has noted, including such 3-D simulations does not detract from the conclusions we have already obtained.

We note that the model could be extended to account for 3-D migration. The equations listed in the Methods section would remain largely unchanged, however the orientation of fibronectin fibers would need to be described with 3-D spherical coordinates rather than 2-D polar coordinates. This would affect the differential equation describing how cells update the fiber orientation. Other agent-based models have considered such 3D polarity rearrangements, though, so this should not present a major obstacle (Germann et al., 2019). The distributions from which one samples the directions of the haptotactic and cell-cell repulsion forces would also have to change, such that we would use a 3D von Mises-Fisher distribution rather than a 2D von Mises distribution. We have noted this lack of 3-D simulations and our plans to address it in future work in the Discussion section.

Lines 104-115 – The authors could consider (I truly mean consider) adding a late-breaking pre-print that is related to the authors' area of study to their already excellent literature review – https://www.biorxiv.org/content/10.1101/2022.11.21.514608v1. I mention it only because it is late breaking, attempting to enable the authors to include should they wish to.

We thank this reviewer for bringing this manuscript to our attention. We have revised our Reference section to include it and have cited it (and its conclusions) in the Discussion section.

Line 150 – The figure "title" does not match the majority of the area dedicated to the figure. Perhaps there could be two figures? Or a different title? "in vivo experiments and results"?

We have changed the figure title to “in vivo observations of Fibronectin and results from gain/loss of function experiments” so that it is more reflective of the content.

Line 152 – Developmental stage in the caption doesn't match the stage identified in Line 141. Should it?

We have clarified in the figure legend that the schematic in Figure 1A is at HH15 and the FN expression data is earlier at HH12.

Figure 2 and Caption (Lines 192-200)Figure 2a and b: I found it difficult to distinguish between velocity vectors and fiber orientation vectors. Could a different color or symbol be considered to more easily distinguish them? Also, I am not seeing the additional puncta laid down by secretory cells. I have assumed that all puncta in the simulation are being visualized in the simulation stills but see only the original grid. Finally, I assume that the extra column of puncta on the right-hand side is just an artifact of visualization or something, but it may be appropriate to address why it is there. Of course, regardless it won't affect the simulations as the cells don't make it that far.

We have added new supplementary figures and movies showing only the evolution of fibronectin puncta and fibers (Figure 2 —figure supplements 3-4; Figure 2 – videos 3-4). The reviewer is correct that the extra column of puncta in the simulations is an artefact of visualization, and we have introduced a sentence in the Figure 2 caption explaining this. As the reviewer has noted, however, the simulations are not affected by this extra column because the cells do not travel that far.

Caption – There appears to be a statistical test indicated in Figure 2C, but the caption does not identify it. The abbreviation "VE" appears in 2D but is not defined in the caption. I think it may not be defined in the manuscript at all, but of course, I may have missed it.

We have changed the abbreviation in Figure 2 from “VE” to “Rep.” to make clear that this refers to parameters relevant to cell-cell repulsion. We have changed this for the other figures as well (Figure s 2, 3, and 5A of the main text are those specifically affected). A two-sample t-test was used to evaluate the statistical significance of Sobol indices by comparing the indices produced from a parameter of interest to those obtained from a dummy parameter. While this test is already mentioned in the “Sensitivity analysis” part of the Methods section, we have also included it in the caption for Figure 2 to improve clarity.

Figure 3 and CaptionVE isn't defined. If it's defined before, this should be fine – I point it out only in case for whatever reason the abbreviation changes in Figure 2, leaving Figure 3 as possibly the first time it's used.

We have changed the abbreviation in Figure 2 from “VE” to “Rep.” to make clear that this refers to model parameters relevant to cell-cell repulsion.

Lines 469-473 – It's not clear which of these new experimental observations are from this work and which are from previous works. I may have misunderstood, but I think they are not all new in this work. Would the authors consider citing the other works here (no doubt there were previously cited but the reader may not recall all of this), enabling the reader to know which was added to this work? Or use some other indicator to identify their most recent findings being presented as a novel in this work?

The experimental observations discussed here refer to the results obtained from Figure 1 of this manuscript. However, the reviewer is correct in noting that they are consistent with previous works concerning neural crest cell migration in other systems. For instance, neural crest cells are well known to secrete matrix metalloproteases and other proteins that serve to remodel the extracellular matrix (Leonard and Taneyhill, 2020). Additionally, knocking out fibronectin expression in mice significantly reduces the number of cardiac NCCs that successfully migrate along target corridors (Wang and Astrof, 2016). Furthermore, in vitro studies of NCC migration suggest that NCCs preferentially orient fibronectin fibers in the direction of their migration and move along dense regions of fibronectin (Newgreen, 1989; Rovasio et al., 1983). We have added a reference to Figure 1 in the Discussion section that makes clear the fact that Lines 469-473 of the original manuscript refer to experiments contained within this manuscript and have added sentences in the Discussion section that highlight the agreement between our model and these observations from other labs.

Line 570 – I believe there may be a typo – I think the computational domain has an area of 250,000 um^2 (500 by 500).

We thank the reviewer for pointing out this typo. We have corrected the text as per the suggestion.

Line 574 – If FN diffused or decayed at a shorter timescale than NCC migration, I feel like those phenomena would have to be accounted for. I may be misreading, but would it be that the FN processes are on a "longer" time scale than NCC migration, thus the FN can be considered unchanged during the 720 minutes of simulated time?

The reviewer is correct that this is a typo, and we should have written “longer time scale than NCC migration”. We have changed the text to reflect this.

General comment on overview – I think it would be reasonable to highlight here how many agents become part of a simulation typically as well as the amount of simulated time to provide readers with an idea of those scales of the simulations. On a similar note, it would be reasonable to note a typical walltime as well as roughly what hardware was used to run the simulations, giving the community a chance to understand how long it might take to run similar simulations.

We have introduced a new paragraph in the “Initial and boundary conditions” part of the Methods section that makes clear that, for our choice of parameters, we initialize seven leader cells and at most 150 follower cells enter the domain (Figure 2 —figure supplement 8). The walltime varies between 90-120 seconds per realization on a workstation composed of four cores (Intel(R) Core(TM) i5-7500T CPU @ 2.70 GHz), depending on how many cells and puncta are added to the domain.

Table 1 – R_cell parameter – I didn't see a McLennan et al. 2020 in the citations. I may have missed it.

We have added the McLennan et al., 2020 reference.

Reviewing the code, I see that there seem to be puncta agents in addition to the fibronectin field. How are the puncta agents used in the simulations? How do they interact with the non-diffusing substrate field? How, if at all, do their properties (their non-coding/PhysiCell-related properties) differ from the original puncta field?

The substrate field to which this reviewer refers is only used to visualize the fibronectin field. During the computations, it is the “puncta agents” that are most important in determining the cell velocities. This is because this “puncta agents” description allowed us to have fibronectin placed in locations that were not restricted to an equally spaced mesh. The puncta agents do not respond to, or exert any, forces; they simply demarcate the locations of fibronectin. The fibronectin field is set equal to 30 μm for mesh voxels in which at least one punctum agent is located, which eases plotting. We have added comments in the source code that makes it clear how the “puncta agents” function in the script and how the fibronectin field is related.

In PhysiCell, there is a default value for the radius of interaction that determines where an agent looks for its neighbors. The default value is 30 um. Roughly, how did the authors address this as they did interaction testing over the longer than 30-um distances for interaction testing that would have occurred during the sensitivity analysis?

In initializing the agents in PhysiCell, we alter the radius of interaction parameter by setting it equal to the filopodial radius, R_filo, so that we could explore interactions that occur at distances greater than 30 um. We also increased the size of the agent mesh in PhysiCell (mechanics_voxel_size in main.cpp, line 112) to ensure that cells in neighboring voxels are not neglected when the interaction radius increases. We have updated the Methods section to make clear that we adjust the radius of interaction when we change R_filo.

In the equations for the gradients (Del S_FN and Del S_cell), I see the motivation – cell center i is compared to all the puncta and cell centers in its vicinity. However, when viewing the expression under the summation, it seems like one would eventually calculate one single scalar, versus a field of scalars over which a gradient could be calculated. I certainly assume that I am not seeing the obvious, but could the authors please provide this reviewer with additional explanation for how or what the gradient is calculated over?

Since the gradient operator is linear, the order of the sum and gradient can be interchanged without changing the result. We have opted to write the equation in this way for conciseness. In the code, however, the computation is performed by first taking the gradient resulting from each scalar, then summing resulting vectors. We have adjusted the equations to make clear that the calculation is performed in this way.